# LiHoF$_4$ as a spin-half non-standard quantum Ising system

Tomer Dollberg[1] and Moshe Schechter[1]

1 Department of Physics, Ben-Gurion University of the Negev, Beer Sheva 84105, Israel

## Abstract

LiHoF$_4$ is a magnetic material known for its Ising-type anisotropy, making it a model system for studying quantum magnetism. However, the theoretical description of LiHoF$_4$ using the quantum Ising model has shown discrepancies in its phase diagram, particularly in the regime dominated by thermal fluctuations. In this study, we investigate the role of off-diagonal dipolar terms in LiHoF$_4$, previously neglected, in determining its properties. We analytically derive the low-energy effective Hamiltonian of LiHoF$_4$, including the off-diagonal dipolar terms perturbatively, both in the absence and presence of a transverse field. Our results encompass the full $B_x - T$ phase diagram, confirming the significance of the off-diagonal dipolar terms in reducing the zero-field critical temperature and determining the critical temperature's dependence on the transverse field. We also highlight the sensitivity of this mechanism to the crystal structure by comparing our calculations with the Fe$_8$ system.

## 1  Introduction

LiHoF$_4$, a magnetic material with remarkable properties, has become a focal point for research seeking to unravel the intricacies of quantum magnetism. Known for its distinct Ising-type anisotropy, LiHoF$_4$ serves as an archetype among anisotropic dipolar magnets, encompassing single-molecule magnets and rare-earth magnetic insulators. These magnets, characterized by a substantial anisotropy barrier and a ground state doublet, provide an intriguing test bed for studying perhaps the simplest model for (quantum) magnetism—the (transverse field) Ising model [1–3]. In addition to their fundamental significance, anisotropic dipolar magnets have garnered attention due to their potential applications in nanomagnets, qubits, and memory bits [4–6]. In these regards, LiHoF$_4$ has emerged as one of the most extensively studied materials in this category, providing insights into diverse phenomena such as quantum tunneling [7], quantum criticality [1,8–10], quantum annealing [11], the effects of disorder [12–17], quantum entanglement [18], the formation and dynamics of magnetic domains [19–21], and high-Q nonlinear dynamics [22]. However, despite being regarded as a quintessential Ising magnet, the persistent discrepancy in its $B_x - T$ phase diagram compared to the predictions of the transverse-field Ising model (TFIM), particularly in the high-temperature, low-field regime dominated by thermal fluctuations, calls into question the adequacy of the theoretical description of LiHoF$_4$ in terms of the TFIM.

Dipolar interactions dominate LiHoF$_4$, and over the years, there has been increasing recognition of the role of the off-diagonal terms of the dipolar interaction in determining the properties of the material, particularly its diluted series, LiHo$_x$Y$_{1-x}$F$_4$, where holmium ions are randomly replaced by non-magnetic yttrium ions. In LiHo$_x$Y$_{1-x}$F$_4$, the interplay of an applied transverse field, random disorder, and the off-diagonal dipolar (ODD) terms has been shown to reduce the symmetry of the system, rendering the TFIM an inadequate model for the system [23–27]. Nevertheless, when studying the pure LiHoF$_4$ system, these ODD terms were thus far usually neglected in deriving an effective low-energy Hamiltonian, citing their smallness and vanishing mean-field (MF) contribution [28,29].

In [30], we suggested that the ODD terms are of quantitative importance in determining the phase diagram and numerically showed that their inclusion in the model results in a notable reduction of the low-field critical temperature. Here we analytically derive the effective low-energy Hamiltonian of the pure LiHoF$_4$ system, perturbatively including the off-diagonal dipolar terms, both in the absence and in the presence of a transverse field. Thus, we find that ODD terms manifest in the effective low-energy model as three-body interactions with an overall anti-ferromagnetic preference. Our results confirm the role of the off-diagonal terms in reducing the critical temperature, and the dependence of the mechanism on the transverse field, thus enabling the derivation of the full phase diagram of LiHoF$_4$ in transverse field. Intriguingly, this mechanism of reduction of the critical temperature due to the contribution of the off-diagonal dipolar terms sensitively depends on the structure of the crystal. To emphasize this point, we repeat our calculations for the Fe$_8$ system and show that in this system, the reduction of $T_c$ is negligible, in agreement with experiment.

## 2 Theory

The dominant interactions in LiHoF$_4$ are magnetic dipolar interactions between the Ho$^{3+}$ ions. Combined with its crystal structure, these result in ferromagnetic order below $T_c = 1.53$ K. Its strong single-ion easy-axis anisotropy results in an Ising-like twofold degenerate non-Kramers ground state and a first excited state separated by an energy gap $\Delta \approx 11.5$ K. Previous works aiming to derive an effective low-energy Hamiltonian did so by directly projecting the full microscopic Hamiltonian onto the low-energy subspace spanned by its two lowest-energy states (the ground state doublet is continuously split by the transverse field $B_x$) [28, 29]. In the absence of an applied transverse field, this approach immediately reduces the dipolar interaction to a strictly longitudinal interaction since the $x$ and $y$ electronic angular momentum operators have vanishing matrix elements between the Ising-like ground states. Only at non-zero transverse fields does this projection include terms proportional to off-diagonal dipolar elements, but in this case, they were neglected due to their smallness compared to the longitudinal interaction and the direct interaction with the transverse field. Thus, this projection results in a simple TFIM Hamiltonian without any contribution from ODD terms.

Here we derive an effective low-energy spin-$\frac{1}{2}$ Hamiltonian that perturbatively includes off-diagonal dipolar terms by applying a Schrieffer-Wolff transformation to the full LiHoF$_4$ Hamiltonian prior to the projection, treating the interaction terms as a perturbation. This approach was used by Chin and Eastham [31], but only for $B_x = 0$ and considering only the lowest excited state. Here we generalize their approach for arbitrary transverse field and include all excited crystal-field states. We also introduce an exchange interaction between nearest neighbors of strength $J_{\text{ex}}$. The Hamiltonian is divided into an unperturbed diagonal part, $H_0$, and a perturbation, $H_T$, which consists of the dipolar and exchange interaction terms, i.e., $H_{\text{full}} = H_0 + H_T$, where

$$H_0 = \sum_i V_c\left(\vec{J}_i\right) + g_L \mu_B B_x \sum_i J_i^x, \tag{1}$$

$$H_T = \frac{1}{2} E_D \sum_{\substack{i \neq j \\ \nu, \mu}} V_{ij}^{\nu\mu} J_i^\nu J_j^\mu + J_{\text{ex}} \sum_{\substack{\langle i,j \rangle \\ \nu}} J_i^\nu J_j^\nu \tag{2}$$

with $\mu, \nu \in \{x, y, z\}$. The hyperfine interaction with the nuclear spins is treated separately, as described later. The dipolar interaction is given by $V_{ij}^{\mu\nu} = \left[\delta^{\mu\nu} r_{ij}^2 - 3\left(\vec{r}_{ij}\right)^\mu \left(\vec{r}_{ij}\right)^\nu\right]/r_{ij}^5$, the nearest neighbor exchange is $J_{\text{ex}} = 1.16$ mK [32], and $k_B E_D = \frac{\mu_0 \mu_B^2 g_L^2}{4\pi}$. The crystal field parameters that comprise $V_c$ are those suggested in Ref. [33], which are identical to Ref. [32] except for a small adjustment of $B_6^4(s)$. The low-energy subspace onto which $H_{\text{full}}$ is projected is that in which all of the ions are in either of their two lowest-energy eigenstates induced by $H_0$; the appropriate projection operator is denoted $P_0$. The Schrieffer-Wolff transformation uses a generator $S$, in terms of which the projected low-energy Hamiltonian is given as [34]

$$H_{\text{eff}} = H_0 P_0 + P_0 H_T P_0 + \frac{1}{2} P_0 [S, H_T] P_0 + \mathcal{O}\left(H_T^3\right). \tag{3}$$

Details of the calculation are found in Appendix A.

## 3 Three-state model in zero-field

For concreteness, we perform the procedure explicitly for $B_x = 0$ while considering only the first excited state. This means that before the Schrieffer-Wolff transformation is applied, all

operators of the full Hamiltonian are truncated so that they only act upon the three lowest-energy states. The two lowest-energy states, which in the absence of a transverse field are designated $|\uparrow\rangle$ and $|\downarrow\rangle$, are chosen such that $\langle\uparrow|J^z|\downarrow\rangle = \langle\downarrow|J^z|\uparrow\rangle = 0$. The operators are then given in matrix form by

$$
V_c = \begin{pmatrix} 0 & & \\ & 0 & \\ & & \Delta \end{pmatrix}; \quad J^z = \begin{pmatrix} -\alpha & & \\ & \alpha & \\ & & 0 \end{pmatrix};
$$

$$
J^x = \begin{pmatrix} 0 & 0 & \rho \\ 0 & 0 & \rho \\ \rho & \rho & 0 \end{pmatrix}; \quad J^y = \begin{pmatrix} 0 & 0 & -i\rho \\ 0 & 0 & i\rho \\ i\rho & -i\rho & 0 \end{pmatrix} \tag{4}
$$

where $\Delta = 11.5\,\text{K}$ and eigenbasis of $V_c$ is chosen such that $\rho = 2.34$ and $\alpha = 5.53$ are real[1]. With these simplifications, we are essentially reproducing the results of Ref. [31], and indeed we obtain an identical result (assuming $J_{\text{ex}} = 0$),

$$
H_{\text{eff}}(B_x = 0) = \frac{\alpha^2}{2} E_D \sum_{i\neq j} V_{ij}^{zz} \sigma_i^z \sigma_j^z + \alpha^2 J_{\text{ex}} \sum_{\langle i,j \rangle} \sigma_i^z \sigma_j^z + \sum_i h_i^\mu \sigma_i^\mu + \sum_{\substack{i\neq j \\ \nu,\mu}} \varepsilon_{ij}^{\mu\nu} \sigma_i^\mu \sigma_j^\nu + H_{3B} \tag{5}
$$

where

$$
\begin{aligned}
H_{3B} = &-\frac{\alpha^2 \rho^2}{\Delta} E_D^2 \sum_{i\neq j\neq k} \left( V_{ij}^{xz} V_{ik}^{xz} + V_{ij}^{yz} V_{ik}^{yz} \right) \sigma_j^z \sigma_k^z \\
&-\frac{\alpha^2 \rho^2}{\Delta} E_D^2 \sum_{i\neq j\neq k} \left( V_{ij}^{xz} V_{ik}^{xz} - V_{ij}^{yz} V_{ik}^{yz} \right) \sigma_i^x \sigma_j^z \sigma_k^z \\
&-\frac{\alpha^2 \rho^2}{\Delta} E_D^2 \sum_{i\neq j\neq k} \left( V_{ij}^{xz} V_{ik}^{yz} + V_{ij}^{yz} V_{ik}^{xz} \right) \sigma_i^y \sigma_j^z \sigma_k^z
\end{aligned} \tag{6}
$$

gathers three-body terms[2]. $\sigma_i^\mu$ are Pauli matrices acting within the two-dimensional low-energy subspace of each ion $i$ between the states $|\uparrow\rangle$ and $|\downarrow\rangle$. Note that an even number of $\sigma^z$ operators in each term of $H_{\text{eff}}$ is required by the time-reversal symmetry of the original Hamiltonian at $B_x = 0$. The full forms of $h_i^\mu$ and $\varepsilon_{ij}^{\mu\nu}$ are found in Appendix A; the fields $h_i^\mu$ all vanish due to symmetry in the undiluted case, and the emergent two-body terms $\varepsilon_{ij}^{\mu\nu}$ are two orders of magnitude smaller than the exchange interaction, which itself is significantly smaller than the dominant longitudinal dipolar interaction. The emergent two-body terms are also short-range interactions by virtue of being proportional to the squared dipolar interaction, and thus $\sim 1/r^6$. In contrast, the three-body terms are of the same order of magnitude as the exchange interaction, and, crucially, they are effectively long-range interactions despite being proportional to a product of dipolar interactions due to the additional summation. Thus, the three-body terms are expected to be the most impactful of the new emergent interactions. As we shall immediately see, only the first term of $H_{3B}$ has a non-vanishing MF contribution, making it pivotal in our analysis. Henceforth, this term will be referred to as the "three-body term." The remaining two terms in $H_{3B}$, involving Pauli $x$ and $y$ operators, do not commute

---

[1]These values differ slightly from those of Ref. [31] due to the use of updated crystal-field parameters, as mentioned in the previous section.

[2]Though the first term in $H_{3B}$ consists of only two operators, it is, in essence, a three-body interaction by virtue of its dependence on the *existence* of spin $i$. This dependence is crucial to explaining the $x$-dependent behavior of the diluted series LiHo$_x$Y$_{1-x}$F$_4$, as expanded upon in Ref. [30].

with the dominant $\sigma_i^z \sigma_j^z$ term in $H_{\text{eff}}(B_x = 0)$. Consequently, these terms are classified as "quantum terms" going forward.

Next, we employ a mean-field approximation by neglecting correlation terms $(\sigma_i^\mu - \langle \sigma_i^\mu \rangle)(\sigma_j^\nu - \langle \sigma_j^\nu \rangle) \approx 0$ for any $\mu, \nu$ and $i \neq j$, and denoting $\langle \sigma_j^\mu \rangle \equiv m_\mu$. In general, we then have three coupled self-consistency equations for $\vec{m}$, but, in the case of zero transverse field, the last two terms in (6) vanish in mean-field due to lattice symmetries, leaving us with the usual Ising self-consistency equation $m_z = \tanh\left(\beta b_i^z\right)$. The exact form of $\vec{b}(\vec{m})$ is given in Appendix B. The critical temperature determined by this equation, with $q = 4$ as the number of nearest neighbors, is

$$
\begin{aligned}
T_c(B_x = 0) = &- \alpha^2 E_D \sum_{j(\neq i)} V_{ij}^{zz} - \alpha^2 q J_{\text{ex}} \\
&- \frac{\rho^4 J_{\text{ex}}}{\Delta}\left(2 E_D \sum_{j \in NN(i)} V_{ij}^{xx} + q J_{\text{ex}}\right) \\
&- \frac{\rho^4}{\Delta} E_D^2 \sum_{j(\neq i)} \left[V_{ij}^{xx} V_{ij}^{yy} - \left(V_{ij}^{xy}\right)^2\right] \\
&+ \frac{4\alpha^2 \rho^2}{\Delta} E_D^2 \sum_{k \neq j(\neq i)} V_{ij}^{xz} V_{jk}^{xz}
\end{aligned}
\tag{7}
$$

where the notation $j \in NN(i)$ indicates that $j$ sums over the $q$ nearest neighbors of spin $i$. The last sum in Eq. (7) above, the result of the first of the three three-body terms in $H_{\text{3B}}$ (6), is the predominant correction to the longitudinal interaction that results in a reduction of $T_c$; the two preceding terms in Eq. (7) are much smaller in value. Fig. 1(a) shows a group of three spins strongly affected by the three-body interaction in such a manner that generates an effective anti-ferromagnetic interaction between two of them. Fig. 1(b) illustrates the spectrum of the three states used in the three-state model.

The (long-range) sums in Eq. (7) are calculated using Ewald's method [35, 36], assuming a needle-shaped domain that, in the non-zero transverse field case to be discussed later, is embedded within a spherical sample whose magnetization in the $x$ direction in response to the transverse field is homogeneous [28]. See further discussion in Appendix B. Thus we find that for the three-state model, the three-body term is responsible for a 5% reduction of the MF critical temperature. Subsequent sections will demonstrate, by incorporating higher excited crystal-field states and employing Monte Carlo simulations, that this three-body term is actually responsible for a majority of the existing discrepancy between MF predictions and experimental observations.

## 4   17-state model in $B_x \geq 0$

As mentioned previously, the Schrieffer-Wolff procedure and subsequent projection can be performed for arbitrary non-zero $B_x$, albeit resulting in an excessively complicated effective Hamiltonian compared to (5). Nevertheless, a mean-field approximation can be applied to it just the same, only in this case, the three self-consistency equations are fully coupled, so, to find $T_c$, they are solved numerically for decreasing $T$ until a phase transition is detected. Details of the procedure are related in Appendix C. Here we use the full, rather than truncated, forms of the operators in $H_{\text{full}}$, again using the Schrieffer–Wolff transformation to decouple (up to second order in $H_T$) the two lowest-energy states from the 15 excited states.

The results on the full phase diagram are shown in Fig. 2; once with the Schrieffer-Wolff transformation applied to the full Hamiltonian in (1) and (2) and once with the Schrieffer-

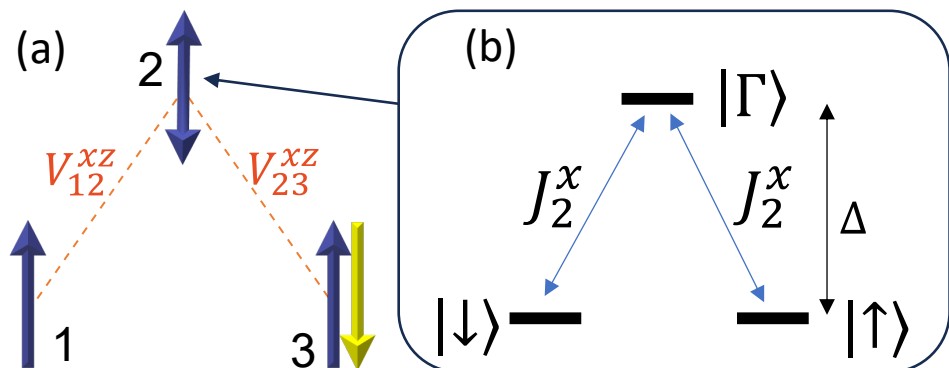

Figure 1: (a) An illustration of three spins, between two of which (denoted 1 and 3) an effective anti-ferromagnetic interaction emerges due to their off-diagonal dipolar interactions with the third (denoted 2). The prominent three-body term in $H_{3B}$ energetically favors a configuration where spin 3 is anti-aligned with spin 1 (yellow), regardless of the state of spin 2 (thus the double-sided arrow), compared to a configuration where spin 3 is aligned with spin 1 (blue). (b) The three lowest-energy levels of $V_c$, used in the three-state model. The existence of an excited level $|\Gamma\rangle$ at spin 2, coupled to the ground states $|\uparrow\rangle$ and $|\downarrow\rangle$ by $J_2^x$, is crucial to the emergence of the three-body interaction, as it modifies the ground state energies of spin 2 in a way that depends on the relative configurations of its neighbors.

Wolff transformation applied to that Hamiltonian but only with $\mu = \nu$. The results presented include two important modifications. First, following Refs. [33, 37, 39], we rescale the longitudinal interaction by 0.805, i.e., we replace $J_i^z J_j^z \rightarrow 0.805 J_i^z J_j^z$ in the original Hamiltonian prior to deriving the effective low-energy Hamiltonian. This is done, as elaborated upon in Appendix E, to account for the strong $c$-axis fluctuations not captured in MF [32]. The factor is chosen slightly larger than suggested in Refs. [33, 37, 39] to match the Monte Carlo results of the current work. Of course, a proper account of fluctuations is warranted where possible and is done by Monte Carlo simulation, as described in the next section. Second, following Ref. [28], we include the effect of the hyperfine interaction with the nuclear spins by assuming its predominant effect is a temperature-dependent renormalization of the transverse magnetic field. Even though the hyperfine interaction has since been studied more comprehensively [8, 23, 40, 41], we opted for this simplified approach since the effect of the hyperfine coupling has not been shown to be significant to the region of the phase diagram of highest interest to this work—near the classical transition [1]. The renormalization process is briefly reproduced in Appendix D.

As previously noted, the complexity of the effective Hamiltonian at $B_x > 0$ precludes a detailed presentation in this section. Nevertheless, we aim to provide insights into the unexpected stability of the ferromagnetic phase under a weak transverse field. Accordingly, we include in Appendix F an extended analysis of the expected behavior of $T_c(B_x = 0)$ (7) at small yet non-zero $B_x$. This analysis reveals that, even when considering only single-ion physics, the influence of the three-body term diminishes with increasing $B_x$. This attenuation partly offsets the expected overall reduction in $T_c$, thereby leading to a steeper rise of the phase boundary.

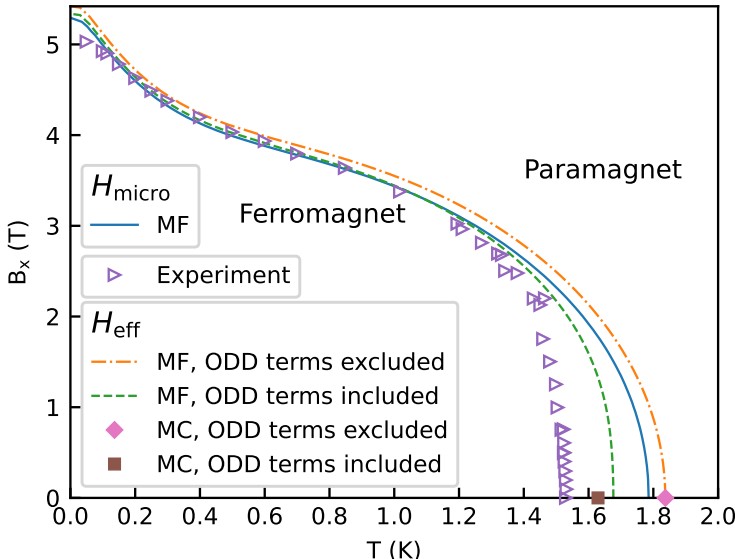

Figure 2: Phase diagram of LiHoF$_4$. The solid blue line is the MF result of Ref. [37], utilizing the full microscopic Hamiltonian, which consists of $H_{\text{full}}$ with the addition of the hyperfine interaction, and with the mean-field rescaled by 0.785. Triangles show experimental results [1,37,38], and all other results are based on the effective Hamiltonian $H_{\text{eff}}$ derived in this work by a Schrieffer-Wolff transformation from the 17-state model. The dot-dashed line represents the MF phase diagram of $H_{\text{eff}}$ derived with the ODD terms explicitly *excluded*, while the dashed line represents MF results with the ODD terms *included*. In both, the longitudinal interaction is scaled by 0.805, as explained in the text. These MF phase diagrams derived from the spin-$\frac{1}{2}$ effective Hamiltonian are corrected for the renormalization of the transverse magnetic field due to the hyperfine interaction, as described in the text. Additionally, results from Monte Carlo simulations at zero-field are represented by a square and a diamond for when ODD terms are included and excluded, respectively. The difference between the two—the result of the three-body term—is in excellent agreement with that found using the more involved MC simulations of Ref. [30].

## 5 Monte Carlo simulations

To properly account for fluctuations we perform classical Monte Carlo (MC) simulations on the effective spin-$\frac{1}{2}$ Hamiltonian obtained from the 17-state model at $B_x = 0$, excluding any quantum terms. Since the quantum terms—i.e., terms involving $\sigma_i^x$ or $\sigma_i^y$—are found to have vanishing mean-field contributions, this simulated Hamiltonian, denoted $H_{\text{MC}}$, is effectively equivalent to that used to generate the MF results in Fig. 2. An explicit form for $H_{\text{MC}}$ is given in Appendix G. Naturally, $H_{\text{MC}}$ does not require any renormalization of the interactions, and, interestingly, the critical temperature of $H_{\text{MC}}$ is found to be slightly smaller than its MF critical temperature, even when the MF includes the scaling factor purported to account for fluctuations—see Fig. 2.

We stress that unlike the simulation in Ref. [30], which involved diagonalization of the full single Ho$^{3+}$ ion, the current simulation is a simple Metropolis MC that works on the effective spin-$\frac{1}{2}$ Hamiltonian $H_{\text{MC}}$. In comparing the $B_x = 0$ results of the two, we observe good agreement, attributing the existing deviations to the adjustment in the crystal-field parameters and the much larger system sizes achieved in this study. This agreement indicates that the quantum

fluctuations that dictate the reduction of $T_c$ are very well captured by a *classical* three-body term, namely the first term in $H_{3B}$ (6).

## 6 Fe$_8$ molecular magnet

Another material considered a good physical magnetic realization of the transverse field Ising model is the molecular magnet Fe$_8$ in crystal form [3,42–46]. Interestingly, it does not show an obvious discrepancy between experimental results and MF calculations [3], as does LiHoF$_4$, raising questions regarding the universality of the effect henceforth described. To address these questions, in this section, we apply the same treatment to Fe$_8$ as we did for LiHoF$_4$.

The Fe$_8$ crystal consists of large molecular clusters, each described by a simple $S = 10$ spin model subject to a strong uniaxial anisotropy described by the crystal field potential $V_{Fe_8} = -DS_z^2 + E\left(S_x^2 - S_y^2\right)$ where $D > E$ [42], situated on a triclinic lattice. Experimentally, in the absence of a transverse magnetic field, Fe$_8$ orders ferromagnetically below a critical temperature of $T_c \approx 0.6\,\text{K}$ [3], in reasonable agreement with theoretical predictions [44,45].

The only non-negligible interaction between the Fe$_8$ molecular clusters is dipolar [45], so in the absence of an external transverse field, the full Hamiltonian is given by

$$\mathcal{H}_{Fe_8} = \sum_i V_{Fe_8} + \frac{\mu_0 \mu_B^2 g^2}{8\pi} \sum_{\substack{i \neq j \\ \nu,\mu}} V_{ij}^{\nu\mu} J_i^\nu J_j^\mu \tag{8}$$

with $g = 2$ [42]. When we apply the Schrieffer-Wolff transformation and project $\mathcal{H}_{Fe_8}$ onto the subspace in which all molecules are within their ground state doublet, we obtain an effective low-energy spin-$\frac{1}{2}$ Hamiltonian. This Hamiltonian, similarly to LiHoF$_4$, consists of longitudinal dipolar interactions, proportional to $V_{ij}^{zz}$, as well as other two-body and three-body terms, proportional to products of other dipolar components. As before, the emergent three-body interactions dictate a reduction of the MF critical temperature; only now, we find that the contribution of all emergent interactions amount to a negligible correction of less than 0.01%—well below experimental error bars. This is perfectly consistent with the stated lack of discrepancy between MF and experiment. Further details of the calculation are provided in Appendix H.

## 7 Conclusion

The discrepancy between theory and experiment regarding the shape of the phase boundary near the classical transition has persisted as an open question over the past two decades [2,28,29,32,38]. In Ref. [30], it has been shown that the inclusion of ODD terms results in a significant reduction of the critical temperature of LiHoF$_4$ at low transverse fields. A simplified argument was given for why one could expect this reduction to be attenuated with an increasing transverse field, possibly leading to the experimentally established steep rise of the phase boundary at small fields, thus offering a potential resolution of the discrepancy. Unfortunately, due to the numerical nature of that work, the range of transverse fields for which an exact numerical value could be given was not large enough to clearly show an attenuation of the described mechanism.

In this work, we pursued an analytical approach that allowed us to describe the full $B_x - T$ phase diagram of LiHoF$_4$ in MF and thus show that the reduction in $T_c$ is indeed attenuated as $B_x$ is increased. As a natural result, the shape of the phase boundary curve obtained from the effective Hamiltonian that includes ODD terms is closer to the experimental curve, showing a similarly steep rise near the classical transition. We also find that the effect of ODD terms

becomes minimal at $B_x \gtrsim 3.5$ T—about the same value where the standard (rescaled) MF and experimental curves merge [37]. This work also provides an explicit analytical expression describing said reduction at $B_x = 0$ (the last term in Eq. (7)), showing that it results from emergent three-body interactions due to ODD terms.

The $T_c$-reducing effect of ODD terms is not fully captured in our MF analysis, as evidenced by the difference between the $B_x = 0$ MC and MF results when ODD terms are included, as seen in Fig. 2. A careful examination of the summation over three-body interactions (that occurs in, e.g., the last term in Eq. (7)) reveals that the mirror symmetry of the crystal guarantees certain cancellations in the homogeneous system assumed in MF; see Appendix B, specifically Fig. 3 and the pursuing discussion, for further details. This observation explains at least part of the difference between the MF and MC zero-field critical temperatures, as a MC simulation, not constrained by the assumption of homogeneity, would arguably manifest a more pronounced effect for the three-body interaction. Regardless, it is noteworthy and highly encouraging that a simple Ising MC simulation agrees reasonably well with the experimental zero-field critical temperature without requiring further special modifications. Furthermore, the current study represents an improvement over the findings of Ref. [30] as it explicitly acknowledges the omission of quantum terms from the effective Hamiltonian, which could account for the remaining discrepancy between the MC simulation and experimental results at $B_x = 0$. Although extending the simulations to $B_x > 0$ would have been desirable, the complexity of the effective Hamiltonian, which in that case contains dozens of non-commuting terms of similar order of magnitude, made any approach beyond MF impractical. Additionally, it should be noted that a further reduction in $T_c$ could be expected due to the fluctuating transverse spin components not considered in this work; the magnitude of such a reduction is yet undetermined [47].

Another interesting feature is the dependence of the mechanism on the crystal structure, made evident by the application of the analysis to the $Fe_8$ system. Qualitatively, the adjustment of $T_c$ due to ODD terms generally applies to any magnetic anisotropic dipolar system. However, we find that the effect can be quite large, as seen in $LiHoF_4$, but may very well be quantitatively negligible, as seen in $Fe_8$; the determining factor being the crystal structure as manifested in MF in sums of the form $\sum_{j \neq k(\neq i)} V_{ij}^{xz} V_{jk}^{xz}$ and $\sum_{j \neq k(\neq i)} V_{ij}^{yz} V_{jk}^{yz}$. In a way, this is not surprising since, as was shown generally by Luttinger and Tisza [48] and later extended specifically to lithium rare-earth tetrafluorides [49], the nature of the ordered phase of a system of dipoles coupled primarily by dipolar interactions is highly sensitive to the geometric arrangement of said dipoles. Thus, the magnitude of the effect of off-diagonal dipolar interactions could vary markedly depending on crystal structure for the same reasons. Perhaps for some lattice configurations, it can also be of opposite sign, i.e., favoring rather than disfavoring ferromagnetic order.

## Acknowledgements

We would like to thank Amnon Aharony and Markus Müller for useful discussions.

**Funding information**   M.S. acknowledges support from the Israel Science Foundation (Grant No. 2300/19).

## A   Derivation of an effective low-energy Hamiltonian

As discussed in Section 2, the Hamiltonian is divided into an unperturbed diagonal part, $H_0$, and a perturbation, $H_T$, given respectively by Eqs. (1) and (2). We designate the two lowest-

energy eigenstates of the single-ion $H_0$ as $|\alpha\rangle$ and $|\beta\rangle$ and define the many-body low-energy subspace through the projection operator $P_0 = \prod_i (|\alpha_i\rangle \langle \alpha_i| + |\beta_i\rangle \langle \beta_i|)$, i.e., it is the subspace in which all of the ions are in one of their two respective low-energy single-ion electronic states. Since $H_T$ consists of two-body operators, it couples the low-energy subspace to either the subspace in which there is a single ion in its excited state or to that in which there are two.

The low-energy Hamiltonian is then given by [34]

$$H_{\text{eff}} = H_0 P_0 + P_0 H_T P_0 + \frac{1}{2} P_0 \left[ S, (H_T)_{od} \right] P_0 + \mathcal{O}\left( H_T^3 \right) \tag{9}$$

where $S$ is called the generator of the transformation and is given by

$$S = \sum_{i,j} \frac{\langle i | (H_T)_{od} | j \rangle}{E_i - E_j} |i\rangle \langle j| .$$

The notation $(.)_{od}$ indicates that only the block-off-diagonal parts of the operator are considered, that is, $(V)_{od} = P_0 V (\mathbb{I} - P_0) + (\mathbb{I} - P_0) V P_0$.

For the given $H_T$, we can write $S$ explicitly using projection operators,

$$S = \frac{E_D}{2} \sum_{i \neq j} \sum_{\mu, \nu} \sideset{}{'}\sum_{\alpha_1, \alpha_2, \beta_1, \beta_2} \frac{V_{ij}^{\nu\mu} P_i^{\alpha_1} J_i^{\mu} P_i^{\beta_1} \otimes P_j^{\alpha_2} J_j^{\nu} P_j^{\beta_2}}{E_{\alpha_1} - E_{\beta_1} + E_{\alpha_2} - E_{\beta_2}}$$
$$+ \frac{J_{\text{ex}}}{2} \sum_{i \neq j} \sum_{\nu} \sideset{}{'}\sum_{\alpha_1, \alpha_2, \beta_1, \beta_2} \delta_{ij,nn} \frac{P_i^{\alpha_1} J_i^{\nu} P_i^{\beta_1} \otimes P_j^{\alpha_2} J_j^{\nu} P_j^{\beta_2}}{E_{\alpha_1} - E_{\beta_1} + E_{\alpha_2} - E_{\beta_2}},$$

where $\delta_{ij,nn}$ is zero except when spin $i$ and $j$ are nearest neighbors, and the single-ion projection operators are defined by

$$P_i^{\alpha} \equiv \prod_{j(\neq i)} \mathbb{1}_j |\alpha_i\rangle \langle \alpha_i| .$$

It should be noted that $\sum_{\alpha} P_i^{\alpha} = \mathbb{I}$ is the identity operator of the complete system. Further, we denote $E_{\alpha}$ as the single-ion eigenenergy corresponding to the single-ion eigenstate $|\alpha\rangle$. The prime on the sums above indicates that we exclude terms for which $\alpha_1, \beta_1, \alpha_2, \beta_2 \leq 2$ or $\alpha_1, \beta_1, \alpha_2, \beta_2 > 2$ or $\alpha_1, \beta_2 > 2 \ \wedge \ \alpha_2, \beta_1 \leq 2$ or $\alpha_1, \beta_2 \leq 2 \ \wedge \ \alpha_2, \beta_1 > 2$ or $\alpha_1, \beta_1 \leq 2 \ \wedge \ \alpha_2, \beta_2 > 2$ or $\alpha_2, \beta_2 \leq 2 \ \wedge \ \alpha_1, \beta_1 > 2$. Thus, only block-off-diagonal elements of $H_T$, i.e., elements that couple states with a different number of spins outside of the low-energy subspace, are included.

The first two terms in (9) are the naive projections of the full Hamiltonian onto the low-energy subspace and are trivially calculated. Therefore, we focus on the third term. In the interest of clarity, we split the expression for $\frac{1}{2}[S, H_T]$ into four parts; one for each of the combinations of dipolar and exchange interactions,

$$\frac{I}{E_D^2} = \sum_{k \neq l} \sum_{i \neq j} \left[ V_{ij}^{\nu\mu} P_i^{\alpha_1} J_i^{\mu} P_i^{\beta_1} P_j^{\alpha_2} J_j^{\nu} P_j^{\beta_2}, V_{kl}^{\sigma\tau} J_k^{\sigma} J_l^{\tau} \right]$$

$$\frac{II}{E_D J_{\text{ex}}} = \frac{1}{3} \sum_{k \neq l} \sum_{i \neq j} \left[ V_{ij}^{\nu\mu} P_i^{\alpha_1} J_i^{\mu} P_i^{\beta_1} P_j^{\alpha_2} J_j^{\nu} P_j^{\beta_2}, \delta_{kl,nn} J_k^{\sigma} J_l^{\sigma} \right]$$

$$\frac{III}{E_D J_{\text{ex}}} = \frac{1}{3} \sum_{k \neq l} \sum_{i \neq j} \left[ \delta_{ij,nn} P_i^{\alpha_1} J_i^{\nu} P_i^{\beta_1} P_j^{\alpha_2} J_j^{\nu} P_j^{\beta_2}, V_{kl}^{\sigma\tau} J_k^{\sigma} J_l^{\tau} \right]$$

$$\frac{IV}{J_{\text{ex}}^2} = \frac{1}{9} \sum_{k \neq l} \sum_{i \neq j} \left[ \delta_{ij,nn} P_i^{\alpha_1} J_i^{\nu} P_i^{\beta_1} P_j^{\alpha_2} J_j^{\nu} P_j^{\beta_2}, \delta_{kl,nn} J_k^{\sigma} J_l^{\sigma} \right]$$

The factor of 1/3 takes care of the extra sum over spatial coordinates below that is only relevant to the dipolar terms. Thus,

$$\frac{1}{2}[S, H_T] = \frac{1}{8} \sum_{\mu, \nu, \sigma, \tau} \sum_{\alpha_1, \alpha_2, \beta_1, \beta_2}' \frac{I + II + III + IV}{E_{\alpha_1} - E_{\beta_1} + E_{\alpha_2} - E_{\beta_2}}.$$

Using the fact that operators acting on different ions commute, we can simplify the four sums and split them into a sum over pairs of ions and a sum over triplets of ions as follows,

$$\frac{I}{E_D^2} = 2 \sum_{i \neq j} V_{ij}^{\nu\mu} V_{ij}^{\sigma\tau} \left\{ P_i^{\alpha_1} J_i^\mu P_i^{\beta_1} J_i^\sigma \left[ P_j^{\alpha_2} J_j^\nu P_j^{\beta_2}, J_j^\tau \right] + \left[ P_j^{\alpha_1} J_j^\nu P_j^{\beta_1}, J_j^\tau \right] J_i^\sigma P_i^{\alpha_2} J_i^\mu P_i^{\beta_2} \right\}$$
$$+ 4 \sum_{i \neq j \neq k} V_{ij}^{\nu\mu} V_{kj}^{\sigma\tau} P_i^{\alpha_1} J_i^\mu P_i^{\beta_1} J_k^\sigma \left[ P_j^{\alpha_2} J_j^\nu P_j^{\beta_2}, J_j^\tau \right]$$

$$\frac{II}{E_D J_{\text{ex}}} = \frac{1}{12} \sum_{i \neq j} V_{ij}^{\nu\mu} \delta_{ij,nn} \left\{ P_i^{\alpha_1} J_i^\mu P_i^{\beta_1} J_i^\sigma \left[ P_j^{\alpha_2} J_j^\nu P_j^{\beta_2}, J_j^\sigma \right] + J_i^\sigma P_i^{\alpha_2} J_i^\nu P_i^{\beta_2} \left[ P_j^{\alpha_1} J_j^\mu P_j^{\beta_1}, J_j^\sigma \right] \right\}$$
$$+ \frac{1}{6} \sum_{i \neq j \neq k} V_{ij}^{\nu\mu} \delta_{kj,nn} P_i^{\alpha_1} J_i^\mu P_i^{\beta_1} J_k^\sigma \left[ P_j^{\alpha_2} J_j^\nu P_j^{\beta_2}, J_j^\sigma \right]$$

$$\frac{III}{E_D J_{\text{ex}}} = \frac{1}{12} \sum_{i \neq j} V_{ij}^{\sigma\tau} \delta_{ij,nn} \left\{ P_i^{\alpha_1} J_i^\nu P_i^{\beta_1} J_i^\sigma \left[ P_j^{\alpha_2} J_j^\nu P_j^{\beta_2}, J_j^\tau \right] + \left[ P_j^{\alpha_1} J_j^\nu P_j^{\beta_1}, J_j^\sigma \right] J_i^\tau P_i^{\alpha_2} J_i^\nu P_i^{\beta_2} \right\}$$
$$+ \frac{1}{6} \sum_{i \neq j \neq k} V_{kj}^{\sigma\tau} \delta_{ij,nn} P_i^{\alpha_1} J_i^\nu P_i^{\beta_1} J_k^\sigma \left[ P_j^{\alpha_2} J_j^\nu P_j^{\beta_2}, J_j^\tau \right]$$

$$\frac{IV}{J_{\text{ex}}^2} = \frac{1}{36} \sum_{i \neq j} \delta_{ij,nn} \left\{ P_i^{\alpha_1} J_i^\nu P_i^{\beta_1} J_i^\sigma \left[ P_j^{\alpha_2} J_j^\nu P_j^{\beta_2}, J_j^\sigma \right] + \left[ P_j^{\alpha_1} J_j^\nu P_j^{\beta_1}, J_j^\sigma \right] J_i^\sigma P_i^{\alpha_2} J_i^\nu P_i^{\beta_2} \right\}$$
$$+ \frac{1}{18} \sum_{i \neq j \neq k} \delta_{ij,nn} \delta_{kj,nn} P_i^{\alpha_1} J_i^\nu P_i^{\beta_1} J_k^\sigma \left[ P_j^{\alpha_2} J_j^\nu P_j^{\beta_2}, J_j^\sigma \right]$$

Since $S$ is block-off-diagonal by construction, its commutator with the block-diagonal part of $H_T$ necessarily lies outside of the low-energy block and would be eliminated by the projection operators, that is, $\frac{1}{2} P_0 [S, (H_T)_d] P_0 = 0$. Therefore $\frac{1}{2} P_0 [S, (H_T)_{od}] P_0 = \frac{1}{2} P_0 [S, H_T] P_0$, and the latter is used in the main text, particularly Eq. (3).

The sums over operators ($\mu$, $\nu$, $\tau$, $\sigma$) and crystal field levels ($\alpha_1, \alpha_2, \beta_1, \beta_2$) are calculated for a given $B_x$ using *Mathematica*.

For the three-state model in zero-field, the above commutators result in $H_{\text{eff}}(B_x = 0)$, as given in Eq. (5). It includes effective fields $h_i^\mu$ and two-body interactions $\varepsilon_{ij}^{\mu\nu}$, which are given here in terms of the parameters $\Delta$, $\rho$, and $\alpha$, described in the main text. Elements not specified are zero; indeed, neither $\varepsilon_{ij}^{xz}$, $\varepsilon_{ij}^{yz}$ nor any term with an odd number of $\sigma^z$ operators can arise at $B_x = 0$ due to time reversal symmetry [31].

$$\varepsilon_{ij}^{xx} = \frac{\rho^4 E_D^2}{4\Delta} \left( 2\left(V_{ij}^{xy}\right)^2 - \left(V_{ij}^{xx}\right)^2 - \left(V_{ij}^{yy}\right)^2 \right)$$

$$- \frac{\rho^4}{2\Delta} J_{\text{ex}} \delta_{ij,nn} \left[ E_D \left( V_{ij}^{xx} + V_{ij}^{yy} \right) + J_{\text{ex}} \right] \tag{10}$$

$$\varepsilon_{ij}^{yy} = -\frac{\rho^4 E_D^2}{2\Delta} \left( \left(V_{ij}^{xy}\right)^2 + V_{ij}^{xx} V_{ij}^{yy} \right)$$

$$- \frac{\rho^4}{2\Delta} J_{\text{ex}} \delta_{ij,nn} \left[ E_D \left( V_{ij}^{xx} + V_{ij}^{yy} \right) + J_{\text{ex}} \right] \tag{11}$$

$$\varepsilon_{ij}^{zz} = \frac{\rho^4 E_D^2}{2\Delta} \left( V_{ij}^{xx} V_{ij}^{yy} - \left(V_{ij}^{xy}\right)^2 \right) \tag{12}$$

$$+ \frac{\rho^4}{2\Delta} J_{\text{ex}} \delta_{ij,nn} \left[ E_D \left( V_{ij}^{xx} + V_{ij}^{yy} \right) + J_{\text{ex}} \right] \tag{13}$$

$$\varepsilon_{ij}^{xy} = \frac{\rho^4 E_D^2}{2\Delta} \left( V_{ij}^{yy} V_{ij}^{xy} - V_{ij}^{xx} V_{ij}^{xy} \right)$$

$$\varepsilon_{ij}^{xy} = \varepsilon_{ij}^{yx}$$

$$h_i^x = \frac{\alpha^2 \rho^2 E_D^2}{\Delta} \sum_{j(\neq i)} \left( \left(V_{ij}^{yz}\right)^2 - \left(V_{ij}^{xz}\right)^2 \right)$$

$$+ \frac{\rho^4 E_D^2}{2\Delta} \sum_{j(\neq i)} \left( \left(V_{ij}^{yy}\right)^2 - \left(V_{ij}^{xx}\right)^2 \right)$$

$$+ \frac{J_{\text{ex}} \rho^4 E_D}{\Delta} \sum_{j(\neq i)} \delta_{ij,nn} \left( V_{ij}^{yy} - V_{ij}^{xx} \right) \tag{14}$$

$$h_i^y = -\frac{2\alpha^2 \rho^2}{\Delta} E_D^2 \sum_{j(\neq i)} \left( V_{ij}^{xz} V_{ij}^{yz} \right)$$

$$- \frac{\rho^4}{\Delta} E_D^2 \sum_{j(\neq i)} V_{ij}^{xy} \left( V_{ij}^{xx} + V_{ij}^{yy} \right)$$

$$- \frac{2 J_{\text{ex}} \rho^4}{\Delta} E_D \sum_{j(\neq i)} \delta_{ij,nn} V_{ij}^{xy} \tag{15}$$

$$h_i^z = 0 \tag{16}$$

These expressions coincide[3] with those given in the appendix to Ref. [31] when $J_{\text{ex}}$ is taken to be zero.

Throughout this section, we derive $H_{\text{eff}}$ up to second order in $H_T$. This cutoff is justified since the relative contribution of terms arising from higher-order corrections is diminished by a factor of $\langle V \rangle / \Delta \sim 1/10$, where $\langle V \rangle$ symbolically represents the energy scale associated with the dipolar interactions. However, it must be acknowledged that higher-order terms also include interactions between a larger number of spins, i.e., four-body terms, five-body terms,

---

[3]Except for a factor of half between the second term in Eq. (15) here and Eq. (23) in Ref. [31], likely due to a misprint. This term vanishes for the pure system, so the difference is inconsequential.

etc. These interactions cannot be naively discounted based solely on the relative smallness of their coefficient. For instance, Rau *et al.* [50] perform a similar perturbation procedure on the pyrochlore XY antiferromagnet $Er_2Ti_2O_7$, and make the case that due to the combinatoric factor associated with four-spin and six-spin terms, their contribution is comparable to that of lower-order terms in the perturbation series. The combinatoric factor is the number of ways an open chain of nearest-neighbor spins can be chosen in the pyrochlore system (since, in that work, only nearest-neighbor interactions are considered). This factor enhances the effect of the four-spin term by two orders of magnitude due to the relatively high coordination number of the pyrochlore system ($q = 6$, compared to $q = 4$ in $LiHoF_4$). In the present case, instead of a combinatoric factor, it is a sum over multi-spin interactions—some of which, as noted in the main text, are long-ranged—that seemingly results in a significant enlargement of the otherwise inherently small higher-order terms.

Despite all of the above, we assert that higher-order terms are negligible in the present case. First, we argue that for an order-of-magnitude estimate, it suffices to consider only nearest neighbors, despite the long-range nature of the interaction. Consider, for example, the three spins depicted in Fig. 1, assuming spin 1 is a nearest-neighbor of spin 2 and spin 3 is a (different) nearest-neighbor of spin 2. The energy associated with such a triplet due to the first three-body term in $H_{3B}$ is $\frac{\alpha^2\rho^2}{\Delta}E_D^2 V_{12}^{xz}V_{23}^{xz} \approx 0.01\,K$. Considering that there are $4 \times 3 = 12$ ways to choose such a chain of neighboring spins in the $LiHoF_4$ crystal, we estimate the energetic contribution to be $\approx 0.1\,K$. This estimation aligns well with the actual mean-field correction found in this work, as given by the last term in Eq.(7), which is $\approx 0.1\,K$. In essence, nearest neighbors dominate the total sum despite the long-range nature of the interactions because the dipolar-derived emergent interactions provide both positive and negative contributions that partially offset each other. This offsetting effect would similarly apply to four-spin (and higher) interactions as well. Now, let us apply the same reasoning to a hypothetical four-spin interaction that emerges from the next order in the Schrieffer-Wolff transformation. The energetic contribution of such a term, accounting for a combinatoric enhancement, would be $4 \times 3 \times 3 \times \alpha^2\rho^4 E_D^3 [V_{12}^{xz}]^3/\Delta^2 \approx 5\,mK$, so its effect on $T_c$ would be negligible in the context of the present work. To conclude, although the omission of higher-order terms is not trivially justified when they involve multi-spin interactions, our analysis demonstrates that the contribution of such terms is indeed negligible in the $LiHoF_4$ system.

## B  Mean-field approximation at $B_x = 0$

We employ a mean-field (MF) approximation by neglecting correlation terms $(\sigma_i^\mu - \langle\sigma_i^\mu\rangle)(\sigma_j^\nu - \langle\sigma_j^\nu\rangle) \approx 0$ for any $\mu, \nu$ and $i \neq j$, and denoting $\langle\sigma_j^\mu\rangle \equiv m_\mu$. The resulting single-body mean-field Hamiltonian is given by

$$H_{\mathrm{MF}} = -\sum_i b_i^\mu \sigma_i^\mu \tag{17}$$

Where the local fields $b_i^\mu$, simplified by employing the $S_4$ symmetry of the crystal about the $Ho^{3+}$ ions and denoting the number of nearest neighbors $q$, are given by

$$b_i^x = \frac{\rho^4 J_{\text{ex}}}{\Delta} m_x \left( 2E_D \sum_{j(\neq i)} \delta_{ij,nn} V_{ij}^{xx} + q J_{\text{ex}} \right)$$
$$+ \frac{\rho^4}{\Delta} E_D^2 m_x \sum_{j(\neq i)} \left[ \left( V_{ij}^{xx} \right)^2 - \left( V_{ij}^{xy} \right)^2 \right]$$
$$- \frac{2\rho^4}{\Delta} E_D^2 m_y \sum_{j(\neq i)} V_{ij}^{yy} V_{ij}^{xy} \tag{18}$$

$$b_i^y = \frac{\rho^4 J_{\text{ex}}}{\Delta} m_y \left( 2E_D \sum_{j(\neq i)} \delta_{ij,nn} V_{ij}^{xx} + q J_{\text{ex}} \right)$$
$$+ \frac{\rho^4}{\Delta} E_D^2 m_y \sum_{j(\neq i)} \left[ \left( V_{ij}^{xy} \right)^2 + V_{ij}^{xx} V_{ij}^{yy} \right]$$
$$- \frac{2\rho^4}{\Delta} E_D^2 m_x \sum_{j(\neq i)} V_{ij}^{yy} V_{ij}^{xy} \tag{19}$$

$$b_i^z = -\alpha^2 m_z E_D \sum_{j(\neq i)} V_{ij}^{zz} - \alpha^2 q J_{\text{ex}} m_z$$
$$- \frac{\rho^4 J_{\text{ex}} m_z}{\Delta} \left( 2E_D \sum_{j(\neq i)} \delta_{ij,nn} V_{ij}^{xx} + q J_{\text{ex}} \right)$$
$$- \frac{\rho^4 m_z}{\Delta} E_D^2 \sum_{j(\neq i)} \left[ V_{ij}^{xx} V_{ij}^{yy} - \left( V_{ij}^{xy} \right)^2 \right]$$
$$+ \frac{4\alpha^2 \rho^2 m_z}{\Delta} E_D^2 \sum_{k \neq j(\neq i)} V_{ij}^{xz} V_{jk}^{xz} \tag{20}$$

The self-consistency equations for a spin-$\frac{1}{2}$ in an arbitrary mean-field $\vec{b}\,(\vec{m})$ at temperature $k_B T = \beta^{-1}$ are [51]
$$m_\mu = \frac{b^\mu}{|\vec{b}|} \tanh\left( \beta \, |\vec{b}| \right)$$
and it is readily observed that $m_x = m_y = 0$ satisfies the first two equations, leaving us with the usual Ising self-consistency equation
$$m_z = \tanh\left( \beta \, b_i^z \right)$$
from which we get $T_c$ given by Eq. (7) in the main text.

**Lattice sums**

In order to get a numerical estimation of $T_c$, we need to calculate the sums in Eq. (7) in the main text and similar sums arising at $B_x > 0$. We define
$$A_{\mu\nu} = \sum_{j(\neq i)} V_{ij}^{\mu\nu}$$
$$B_{\mu\nu,\sigma\tau} = \sum_{j(\neq i)} V_{ij}^{\mu\nu} V_{ij}^{\sigma\tau}$$

and express the various sums that appear in the mean fields derived from either the three-state or 17-state models using these definitions,

$$
\begin{aligned}
\sum_{j \neq k(\neq i)} V_{ki}^{\mu\nu} V_{kj}^{\sigma\tau} &= \sum_{k(\neq i)} V_{ki}^{\mu\nu} \left( \sum_{j(\neq k \neq i)} V_{kj}^{\sigma\tau} \right) \\
&= \sum_{k(\neq i)} V_{ki}^{\mu\nu} \left( -V_{ki}^{\sigma\tau} + \sum_{j(\neq k)} V_{kj}^{\sigma\tau} \right) \\
&= \sum_{k(\neq i)} V_{ki}^{\mu\nu} \left( -V_{ki}^{\sigma\tau} + A_{\sigma\tau} \right) \\
&= A_{\sigma\tau} A_{\mu\nu} - B_{\mu\nu,\sigma\tau}.
\end{aligned}
\tag{21}
$$

Similarly, we also have $\sum_{j \neq k(\neq i)} V_{ij}^{\mu\nu} V_{ik}^{\sigma\tau} = A_{\sigma\tau} A_{\mu\nu} - B_{\mu\nu,\sigma\tau}$.

As the summand in $B_{\mu\nu,\sigma\tau}$ is $\sim 1/r^6$, it converges absolutely and rapidly, and is thus best numerically evaluated directly (values summarized in Table 1). The sum $A_{\mu\nu}$ is a bit more subtle; since it is $\sim 1/r^3$, it is conditionally convergent, meaning its value depends on the summation order—that is, it depends on the limiting sample shape.

Magnetic domains forming below $T_c$ in a pattern of long thin cylinders (or needles) parallel to the easy axis [52,53] are understood as the way the system minimizes stray fields due to their energetic cost [54]. This is the justification for the use of an infinitely long thin cylinder as the limiting shape in the mean-field approximation when calculating $A_{zz}$ [28, 29, 55, 56]. Nonetheless, as noted by Chakraborty *et al.* [28], the transverse component of the magnetization, which develops in response to the applied transverse field, is indifferent to this domain structure. Therefore, the shape-dependent transverse dipolar interactions, $A_{xx}$ and $A_{yy}$, should be calculated by summing over the entire sample and not just a single needle-shaped domain as with $A_{zz}$. Assuming the principal axes of the sample are aligned with the single-ion magnetic axes, the off-diagonal sums vanish, so $A_{\mu\nu} = 0$ for $\mu \neq \nu$.

To calculate these sums $A_{\mu\mu}$, we use Ewald's method to obtain their Fourier transform and evaluate it at $\vec{k} \to 0$. The Fourier transform of the dipolar interaction for a lattice with a basis is given by [36]

$$
\begin{aligned}
V_{\vec{k}}^{\alpha\beta} = &\sum_{\vec{\tau}} \frac{4\pi}{v} \frac{k^\alpha k^\beta}{k^2} \exp\left(-k^2/\left(4R^2\right)\right) \\
&+ \frac{4\pi}{v} \sum_{\vec{\tau}} \sum_{\vec{G} \neq 0} \frac{\left(G^\alpha + k^\alpha\right)\left(G^\beta + k^\beta\right)}{\left|\vec{G} + \vec{k}\right|^2} \exp\left(-\left|\vec{G} + \vec{k}\right|^2 / \left(4R^2\right) - i\vec{G} \cdot \vec{\tau}\right) \\
&- {\sum_{\vec{\tau}}}' \sum_{\vec{r}_l} \left\{ \frac{2R \exp\left(-R^2 \left(\vec{r}_l + \vec{\tau}\right)^2\right)}{\sqrt{\pi}\left(\vec{r}_l + \vec{\tau}\right)^2} \left[ \left(3 + 2R^2 \left(\vec{r}_l + \vec{\tau}\right)^2\right) \frac{\left(r_l^\alpha + \tau^\alpha\right)\left(r_l^\beta + \tau^\beta\right)}{\left(\vec{r}_l + \vec{\tau}\right)^2} - \delta_{\alpha\beta} \right] \right. \\
&\left. + \left( \frac{3\left(r_l^\alpha + \tau^\alpha\right)\left(r_l^\beta + \tau^\beta\right)}{\left(\vec{r}_l + \vec{\tau}\right)^5} - \frac{\delta_{\alpha\beta}}{\left(\vec{r}_l + \vec{\tau}\right)^3} \right) \operatorname{erfc}\left(R\left|\vec{r}_l + \vec{\tau}\right|\right) \right\} \exp\left(-\vec{k} \cdot \vec{r}_l\right) - \frac{4R^3}{3\sqrt{\pi}} \delta_{\alpha\beta}
\end{aligned}
\tag{22}
$$

where $\vec{\tau}$ are the vectors indicating the locations of atoms in the basis, $\vec{G}$ are vectors of the reciprocal lattice, and $\vec{r}_l$ are vectors of the real underlying Bravais lattice. $v$ is the volume of a unit cell, and $R$ is a parameter used to balance the convergence of the real and reciprocal sums, usually set to $R = \frac{2}{a}$. The prime on the sum indicates that the origin is excluded, i.e., $\vec{\tau} \neq 0$ when $\vec{r}_l = 0$. It is important to note that the first term in (22) is non-analytic as $\vec{k} \to 0$—a

manifestation of the shape dependence discussed above. From a macroscopic point of view, the shape dependence of the sums $A_{\mu\nu}$ translates to a shape-dependent demagnetization factor. Defining the demagnetization factors as [57]

$$L_\mu = 4\pi \lim_{k\to 0} \left( \frac{k_\mu}{k} \right)^2$$

and using the longitudinal factor $L_z = 0$ (appropriate to a needle-shaped domain) and the transverse factors $L_x = L_y = \frac{4\pi}{3}$ (appropriate to a spherical sample) [54], we obtain

$$A_{zz} = V_{\vec{k}=0}^{zz} = -11.271 \left[ a^{-3} \right] \tag{23}$$

$$A_{xx} = A_{yy} = V_{\vec{k}=0}^{xx} = 1.603 \left[ a^{-3} \right] \tag{24}$$

where distances are measured in units of $a = 5.175\,\text{Å}$. These are consistent with the values in Ref. [32], subtracting the respective demagnetization factors.

Table 1: Values of $B_{\mu\nu,\sigma\tau} = \sum_{j(\neq i)} V_{ij}^{\mu\nu} V_{ij}^{\sigma\tau}$ calculated by direct summation; values not specified are zero. $a = 5.175\,\text{Å}$ is the length of the LiHoF$_4$ unit cell.

| | |
|---|---|
| $B_{zz,zz}$ | $17.93\,a^{-6}$ |
| $B_{xx,yy},B_{yy,xx}$ | $-22.05\,a^{-6}$ |
| $B_{yy,yy},B_{xx,xx}$ | $31.02\,a^{-6}$ |
| $B_{xz,xz},B_{xz,zx}, B_{zx,xz},B_{zx,zx},$ $B_{yz,yz},B_{yz,zy}, B_{zy,yz},B_{zy,zy}$ | $36.73\,a^{-6}$ |
| $B_{xy,xy},B_{xy,yx}, B_{yx,xy},B_{yx,yx}$ | $4.80\,a^{-6}$ |
| $B_{yy,zz},B_{zz,yy}, B_{xx,zz},B_{zz,xx}$ | $-8.96\,a^{-6}$ |

Sums of the form $\sum_{j(\neq i)} V_{ij}^{\mu\nu} \delta_{ij,nn}$ are trivially calculated by summing the four dipolar terms associated with the four nearest neighbors.

In the case of $B_x > 0$, we also encounter slightly more subtle sums of the form $\sum_{j\neq k(\neq i)} V_{ij}^{\mu\nu} \delta_{jk,nn}$, which can be expressed as

$$\sum_{j\neq k(\neq i)} V_{ij}^{\mu\nu} \delta_{jk,nn} = q \sum_{j(\neq i)} V_{ij}^{\mu\nu} - \sum_{j(\neq i)} \delta_{ij,nn} V_{ij}^{\mu\nu}.$$

Eventually, combining Eq. (21), the values listed in Table 1, and the values given in Eqs. (23) and (24), we can evaluate Eq. (7) from the main text to find $T_c = 2.18\,\text{K}$, which is about 5% lower than the critical temperature found from applying a mean-field approximation directly to the microscopic Hamiltonian ($T_c = 2.27\,\text{K}$).

An interesting consequence of Eq. (21) is that the three-body sum in Eq. (7) in the main text, $\sum_{k\neq j(\neq i)} V_{ij}^{xz} V_{jk}^{xz}$, is directly proportional to $B_{xz,xz}$. This is a due to the mirror symmetry through a plane parallel to the $y-z$ plane which contains spin $j$. As illustrated in Fig. 3, this symmetry guarantees that a contribution to the field at spin $i$, by any spin $k$ located at $\vec{r}_{ij} + (x_{jk}, y_{jk}, z_{jk})$ is offset by another spin $k$ located at $\vec{r}_{ij} + (-x_{jk}, y_{jk}, z_{jk})$. The only contribution not reduced by this symmetry comes from the spin $k$ that is located at $\vec{r}_{ij} + (x_{ij}, -y_{ij}, -z_{ij}) = (2x_{ij}, 0, 0)$, since its counterpart is omitted from the sum by the restriction $i \neq k$. Consequently, a MF approximation only considers three-body interactions where the third spin is an equal distance, along the $x$-axis, from the two other spins. This explains why a MC simulation shows a greater reduction of $T_c$ than seen in MF (see Fig. 2 in the main text); in MC not all spins are in the same state, so cancellations due to the mirror symmetry are not guaranteed.

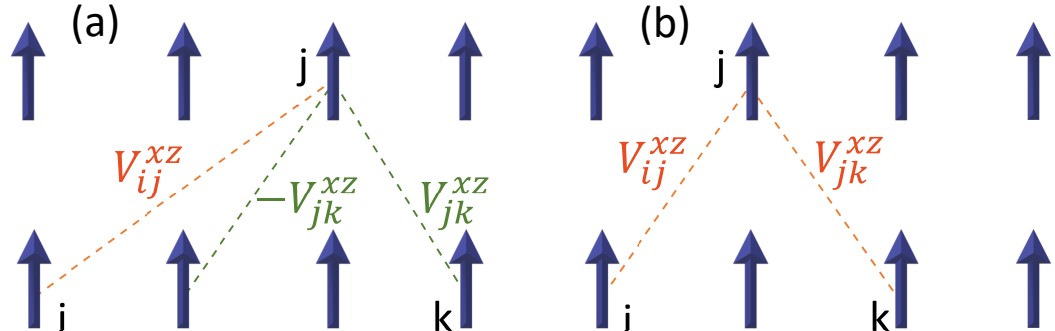

Figure 3: An illustration demonstrating the cancellations occurring in mean-field. (a) For a given pair of spins $i$ and $j$, the three-body interaction involving a typical third spin $k$, $V_{ij}^{xz}V_{jk}^{xz}$, would tend to offset by the three-body interaction involving the mirror image of $k$ relative to $j$ (indicated by $-V_{jk}^{xz}$). (b) The exception where the interactions do not cancel is when $k$ is chosen such that its mirror image is $V_{ij}^{xz}$, which is omitted from the sum by the restriction $k \neq i$. The cancellations are only guaranteed in MF, where the system is assumed homogeneous, but not in a MC simulation or, indeed, the physical system.

## C  Mean-field approximation at $B_x \geq 0$

As mentioned in the main text, the procedure described in Appendix A can be performed for arbitrary non-zero $B_x$. When a mean-field approximation is applied to the resultant effective Hamiltonian, the three resulting self-consistency equations are fully coupled. Thus, to find $T_c$, the self-consistency equations are solved numerically for decreasing $T$ until a non-zero value for $\langle \sigma^z \rangle$ is detected. We note that, at $B_x > 0$, the low-energy subspace is spanned by the two lowest-energy eigenstates of $H_{\text{single-site}} = V_c(\vec{J}) - g_L \mu_B B_x J^x$, denoted $|\alpha\rangle$ and $|\beta\rangle$. In this case, we are no longer interested simply in $\langle \sigma^z \rangle \equiv m_z$ as it no longer approximates magnetization along the physical $z$-axis. Instead, for each value $B_x$, we decompose the two-dimensional projection of $J^\mu$, as done in Ref. [28], defining

$$J_{\text{eff}}^\mu := (|\alpha\rangle \langle \alpha| + |\beta\rangle \langle \beta|) J^\mu (|\alpha\rangle \langle \alpha| + |\beta\rangle \langle \beta|) \mapsto C_\mu + \sum_{\nu=x,y,z} C_{\mu\nu}(B_x)\sigma^\nu. \qquad (25)$$

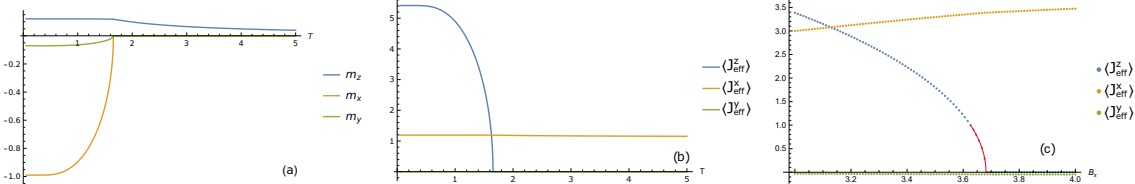

Figure 4: Magnetization vs. temperature and transverse field. (a) Raw solutions of the self-consistency equations, $m_x$, $m_y$, and $m_z$, as a function of temperature at $B_x = 1\,\text{T}$; (b) mean values of the composite quantities $J_{\text{eff}}^x$, $J_{\text{eff}}^y$, and $J_{\text{eff}}^z$, found from the raw solutions at $B_x = 1\,\text{T}$ by (25). (c) Mean values of $J_{\text{eff}}^\mu$ as a function of $B_x$ at $T = 0$. The solid red line is a best-fit of $\langle J_{\text{eff}}^z \rangle \propto \sqrt{B_x^c - B_x}$. Due to the relative numerical complexity, the latter approach is used only at $T \leq 0.5\,\text{K}$.

Then, after self-consistently obtaining $m_x$, $m_y$, and $m_z$, we can plot $\langle J_{\text{eff}}^\mu \rangle = C_{\mu z}m_z + C_{\mu y}m_y + C_{\mu x}m_x + C_\mu$ vs. temperature. $\langle J_{\text{eff}}^z \rangle$, in particular, acts as an

order parameter allowing for the identification of $T_c$. Figs. 4(a) and 4(b) compare $m_\mu$ with $\langle J_{\text{eff}}^\mu \rangle$ for $B_x = 1\,\text{T}$.

It might seem initially surprising that $\langle J_{\text{eff}}^x \rangle$ is almost independent of $T$, also at $T > T_c$, as seen in Fig. 4(b). This behavior is expected in the ferromagnetic phase of the transverse-field Ising model, but as $T \to \infty$, all expectation values should tend towards zero (as seen in Fig. 4(a)). This apparent issue is resolved by recognizing that the decomposition (25) merely approximates the thermal averages and is only valid at low temperatures. To reproduce the decline and eventual vanishing of $\langle J^x \rangle$, higher excited crystal-field levels must be included in the analysis.

At the low-temperature regime, the critical field is nearly independent of temperature (when the hyperfine interaction is not considered). Therefore, at low temperatures, instead of $\langle J_{\text{eff}}^\mu \rangle$ vs. temperature, we calculate $\langle J_{\text{eff}}^\mu \rangle$ for various $B_x$ at a given temperature and fit $\langle J_{\text{eff}}^z \rangle \propto \sqrt{B_x^c - B_x}$ to find the critical field $B_x^c$. This procedure is illustrated in Fig. 4(c).

## D The hyperfine interaction

In this section, we briefly recount the temperature-dependent renormalization procedure introduced in Ref. [28] to account for the effect of the hyperfine coupling between electronic ($J = 8$) and nuclear ($I = 7/2$) spins of the Ho ions in LiHoF$_4$. The hyperfine coupling is given by a term in the full microscopic Hamiltonian of the form $H_{\text{hf}} = A \sum_i (\boldsymbol{I}_i \cdot \boldsymbol{J}_i)$, with $A = 0.039\,\text{K}$ [7]. While the interaction of electronic spins with their nuclear counterparts in LiHoF$_4$ has attracted significant interest in its own right [8, 10, 23, 40, 41, 58], the procedure we implement in this work approximates its effect as mainly a temperature-dependent renormalization of the transverse field. The procedure is performed following [28]. We consider a single Ho ion subject to the LiHoF$_4$ crystal-field potential and a transverse field $B_x$, and include both its electronic spin $\vec{J}$, its nuclear spin $\vec{I}$ and their mutual interaction. The Hamiltonian of this system is given by

$$H_{\text{hyp}} = \left[ V_c(\vec{J}) - g_L \mu_B B_x J^x \right] \otimes \mathbb{1}_N + A \sum_{\mu=x,y,z} J^\mu \otimes I^\mu, \qquad (26)$$

where $\mathbb{1}_N$ is the identity operator in the Hilbert space of the nuclear spin. $H_{\text{hyp}}$ acts on the Hilbert space that is a tensor product of the (17-dimensional) electronic and (eight-dimensional) nuclear spin Hilbert spaces and is thus of dimension $17 \times 8 = 136$. We denote the eigenstates of the Hamiltonian (26) by $\left| \psi_n^{(\text{no-})\text{hf}} \right\rangle$ and the corresponding eigen-energies by $E_n^{(\text{no-})\text{hf}}$ for the case with (without) the hyperfine interaction, i.e., $A = 0.039\,\text{K}$ ($A = 0$). The single-ion susceptibility can be calculated for a given inverse temperature $\beta \equiv 1/T$ by [59]

$$
\begin{aligned}
\chi_{zz}^{(\text{no-})\text{hf}} = &- \frac{2}{Z_{(\text{no-})\text{hf}}} \sum_{\substack{m,n=1,\dots,136 \\ E_m \neq E_n}}^{E_m \neq E_n} \frac{\left| \left\langle \psi_m^{(\text{no-})\text{hf}} \right| J^z \otimes \mathbb{1}_N \left| \psi_n^{(\text{no-})\text{hf}} \right\rangle \right|^2}{E_n^{(\text{no-})\text{hf}} - E_m^{(\text{no-})\text{hf}}} e^{-\beta E_n^{(\text{no-})\text{hf}}} \\
&+ \frac{\beta}{Z_{(\text{no-})\text{hf}}} \sum_{\substack{m,n=1,\dots,136 \\ E_m = E_n}}^{E_m = E_n} \left| \left\langle \psi_m^{(\text{no-})\text{hf}} \right| J^z \otimes \mathbb{1}_N \left| \psi_n^{(\text{no-})\text{hf}} \right\rangle \right|^2 e^{-\beta E_n^{(\text{no-})\text{hf}}} \\
&- \beta \left[ \sum_{n=1}^{136} \left\langle \psi_n^{(\text{no-})\text{hf}} \right| J^z \otimes \mathbb{1}_N \left| \psi_n^{(\text{no-})\text{hf}} \right\rangle \frac{e^{-\beta E_n^{(\text{no-})\text{hf}}}}{Z_{(\text{no-})\text{hf}}} \right]^2. \qquad (27)
\end{aligned}
$$

Here, $Z_{(\text{no-})\text{hf}} = \sum_{n=1}^{136} e^{-\beta E_n^{(\text{no-})\text{hf}}}$ is the partition function with (without) the hyperfine interaction. $\chi_{zz}^{(\text{no-})\text{hf}}$ is implicitly a function of transverse field $B_x$ through the states $\left| \psi_n^{(\text{no-})\text{hf}} \right\rangle$ and energies $E_n^{(\text{no-})\text{hf}}$. The first sum, known as the Van Vleck contribution, is over states non-degenerate

in energy, while the second sum, known as the Curie contribution, is over states degenerate in energy. The last term includes the squared thermal average of $J^z \otimes \mathbb{1}_N$, which always vanishes in the absence of an applied *longitudinal* magnetic field, as assumed in this appendix.

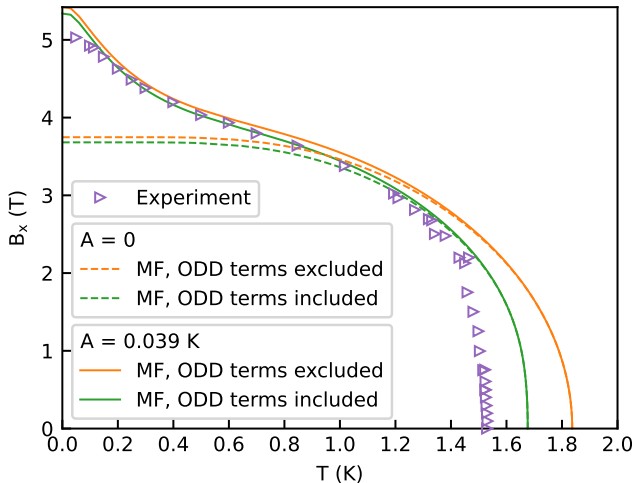

Figure 5: Phase diagram of LiHoF$_4$ showcasing the effect of the temperature-dependent renormalization of the transverse field intended to account for hyperfine interactions. The dashed lines show MF results derived from $H_{\text{eff}}$ without any modification and thereby include no hyperfine interactions ($A = 0$). The solid lines show these results following the application of the renormalization described in the text (corresponding to a hyperfine interaction $A = 0.039$ K) and are the same results shown in Fig. 2

The phase boundary without hyperfine interactions is the collection of points $\left(T_c, B_{x,c}^{\text{no-hf}}\right)$, and is obtained from a MF approximation applied to $H_{\text{eff}}$, as described in the main text. The renormalization of the transverse field due to hyperfine interactions is then taken as a mapping $\left(T_c, B_{x,c}^{\text{no-hf}}\right) \rightarrow \left(T_c, B_{x,c}^{\text{hf}}\right)$. We find $B_{x,c}^{\text{hf}}$ by requiring $\chi_{zz}^{\text{no-hf}}\left(T_c, B_{x,c}^{\text{no-hf}}\right) = \chi_{zz}^{\text{hf}}\left(T_c, B_{x,c}^{\text{hf}}\right)$. The procedure is visually illustrated in Fig. 6 of Ref. [28]. Fig. 5 shows the phase boundary before and after the renormalization procedure described above is applied, in order to illustrate its effect.

## E   Rescaling of the longitudinal interaction

Mean-field theory, applied extensively in this paper, is known to inherently overestimate the values of the critical temperature and critical field due to its neglect of fluctuations. In the case of LiHoF$_4$, the critical values obtained by straightforward application of MF theory to the microscopic Hamiltonian ($H_{\text{full}}$ with the addition of the hyperfine interaction $H_{\text{hf}}$) are $T_c = 2.27$ K and $B_{x,c} = 6.38$ T; both, as expected, significantly higher than their respective experimental values. Ref. [32] addresses this issue by employing a high-density $1/z$ expansion (z is the coordination number), within a so-called *effective medium approach* which considers corrections to MF theory by accounting for single-site fluctuations [60]. Within this framework, the effect of fluctuations, to first order in $(1/z)$, is found to be a rescaling of the longitudinal ($cc$) interaction. Ref. [32] explicitly lists the value of this factor at the critical points ($T_c, B_x = 0$) and $\left(T = 0, B_{x,c}\right)$, and also notes that it decreases "roughly linearly" with temperature. The values given at the two critical points are $(1.3004)^{-1} = 0.769$ and $(1.00493)^{-1} = 0.995$, respectively.

In later works by some of the same authors [37, 39], they opt for a simpler approach, wherein they apply a constant rescaling factor of 0.785 in combination with an adjustment of one crystal-field parameter, $B_6^4(s)$.

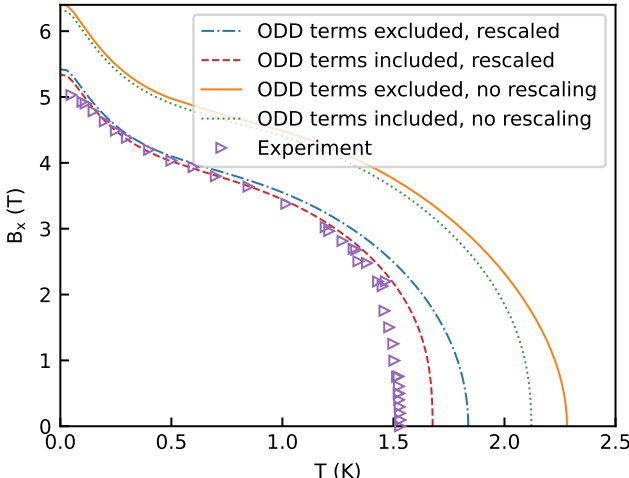

Figure 6: Comparison of $B_x - T$ phase diagrams: Mean-field results for LiHoF$_4$ with the inclusion and exclusion of ODD terms, alongside experimental data. The diagrams illustrate the effects of rescaling the longitudinal interactions, showing that the qualitative features arising from inclusion of ODD terms are preserved with and without rescaling.

In this work, we follow the same approach, using the same crystal-field parameter values, but further adjusting the rescaling factor to 0.805. The motivation is to fit the MF zero-field $T_c$ to the MC result, which fully accounts for fluctuations. The rescaling, $J_i^z J_j^z \to 0.805 J_i^z J_j^z$, is applied directly to the full Hamiltonian before the derivation of the effective Hamiltonian. For comparison, the phase diagram without the rescaling factor is presented in Fig. 6.

## F  Extended analytical characterization of $T_c$ in non-zero $B_x$

Fig. 2 presents the full phase diagram at $B_x \geq 0$. The derivation of this diagram is partly numerical, precluding an analytical expression for $T_c(B_x)$ at $B_x > 0$. To nevertheless further illuminate the different factors responsible for the apparent weak dependence of $T_c$ on $B_x$, we extend the expression for $T_c(B_x = 0)$, given in Eq. (7), to small non-zero $B_x$ values. This extension is done by promoting the parameters $\alpha$, $\rho$, and $\Delta$ to functions dependent on $B_x$. For this purpose, we numerically diagonalize the single-site Hamiltonian,

$$H_{\text{single-site}} = V_c(\vec{J}) - g_L \mu_B B_x J^x, \tag{28}$$

and obtain its three lowest-energy eigenstates, which we denote $|\alpha(B_x)\rangle$, $|\beta(B_x)\rangle$, and $|\Gamma(B_x)\rangle$. Following the methodology of Ref. [28], we then define the states $|\uparrow(B_x)\rangle$ and $|\downarrow(B_x)\rangle$ through a unitary rotation of $|\alpha(B_x)\rangle$ and $|\beta(B_x)\rangle$, ensuring that the matrix elements of $J^z$ between $|\uparrow(B_x)\rangle$ and $|\downarrow(B_x)\rangle$ are real and diagonal for each $B_x$. Given these two states, we define the $B_x$-dependent parameters below

$$\alpha(B_x) := \langle \uparrow(B_x)| J^z |\uparrow(B_x)\rangle \tag{29a}$$

$$\rho_x(B_x) := |\langle \downarrow(B_x)| J^x |\Gamma(B_x)\rangle| \equiv |\langle \uparrow(B_x)| J^x |\Gamma(B_x)\rangle| \tag{29b}$$

$$\rho_y(B_x) := |\langle \downarrow(B_x)| J^y |\Gamma(B_x)\rangle| \equiv |\langle \uparrow(B_x)| J^y |\Gamma(B_x)\rangle| \tag{29c}$$

$$\Delta(B_x) := \langle \Gamma(B_x)| H_{\text{single-site}} |\Gamma(B_x)\rangle$$
$$- \frac{1}{2}\left( \langle \uparrow(B_x)| H_{\text{single-site}} |\uparrow(B_x)\rangle + \langle \downarrow(B_x)| H_{\text{single-site}} |\downarrow(B_x)\rangle \right) \tag{29d}$$

The values of the above parameters are plotted in Fig. 7(b) below for $0 \le B_x \le 3\,\text{T}$. By definition, we have $\rho_x(0) = \rho_y(0)$ and the above definitions become equivalent to the definitions in Eq. (4) at $B_x = 0$. We note that $\rho_x$ and $\rho_y$ are defined separately to account for the symmetry-breaking $B_x$ applied along the $x$ axis. Before we proceed, we must first acknowledge that the expression for $T_c(B_x = 0)$ (7) was derived assuming $\rho_x = \rho_y$, which is no longer valid at $B_x > 0$. Therefore, we re-derive it, at $B_x = 0$ but assuming $\rho_x \ne \rho_y$, and get

$$T_c(B_x) = -\alpha^2 E_D \sum_{j(\ne i)} V_{ij}^{zz} - \alpha^2 q J_{\text{ex}}$$
$$- \frac{\rho_x^2 \rho_y^2 J_{\text{ex}}}{\Delta}\left( 2E_D \sum_{j \in NN(i)} V_{ij}^{xx} + q J_{\text{ex}} \right)$$
$$- \frac{\rho_x^2 \rho_y^2}{\Delta} E_D^2 \sum_{j(\ne i)} \left[ V_{ij}^{xx} V_{ij}^{yy} - \left( V_{ij}^{xy} \right)^2 \right]$$
$$+ \frac{2\alpha^2 \rho_x^2}{\Delta} E_D^2 \sum_{k \ne j(\ne i)} V_{ij}^{xz} V_{jk}^{xz} + \frac{2\alpha^2 \rho_y^2}{\Delta} E_D^2 \sum_{k \ne j(\ne i)} V_{ij}^{yz} V_{jk}^{yz}. \tag{30}$$

The explicit dependence of $\alpha$, $\rho_x$, $\rho_y$, and $\Delta$ on $B_x$ has been omitted for brevity. The above expression is not an exact analytical solution, but rather a heuristic extension of the analytical (MF) $B_x = 0$ form, for two reasons. First, the effective Hamiltonian, derived at $B_x > 0$, involves many more terms that do not appear at $B_x = 0$ and are absent from $T_c(B_x = 0)$. Second, even considering only the $B_x = 0$ effective Hamiltonian, the original expression (7) is contingent upon $m_x = m_y = 0$ being a solution to the self-consistent equations, as mentioned in Appendix B. This is of course no longer the case when a transverse field is applied, leading to non-zero $m_x$.

Still, we expect that at small transverse fields, the above expression is useful for understanding the $T_c(B_x)$ behavior seen in Fig. 2. Indeed, when we compare the difference in $T_c$ caused by the inclusion of off-diagonal dipolar terms, as defined by

$$\Delta T_c(B_x) = -\frac{\rho_x^2 \rho_y^2}{\Delta} E_D^2 \sum_{j(\ne i)} \left( V_{ij}^{xy} \right)^2 - \frac{2\alpha^2 \rho_x^2}{\Delta} E_D^2 \sum_{k \ne j(\ne i)} V_{ij}^{xz} V_{jk}^{xz} - \frac{2\alpha^2 \rho_y^2}{\Delta} E_D^2 \sum_{k \ne j(\ne i)} V_{ij}^{yz} V_{jk}^{yz} \tag{31}$$

to the numerical value, we find adequate agreement up to $B_x \approx 1\,\text{T}$, as seen in Fig. 7(a). Here, the numerical value is the difference between $T_c$ derived with and without off-diagonal dipolar terms in a three-state model. This difference can be seen in Fig. 8 as the difference between the curve labeled *MF, ODD terms excluded* and the curve labeled *3 states*. The fact that this difference decreases with increasing $B_x$ is the key to the steep rise of the phase boundary near the classical critical point. Namely, that steep rise is the result of a balance between, on the one hand, the tendency of $T_c$ to decrease with $B_x$ due to the reduction of the first two terms in Eq. (30), and, on the other hand, the decline of the $T_c$-reducing terms that constitute $\Delta T_c(B_x)$.

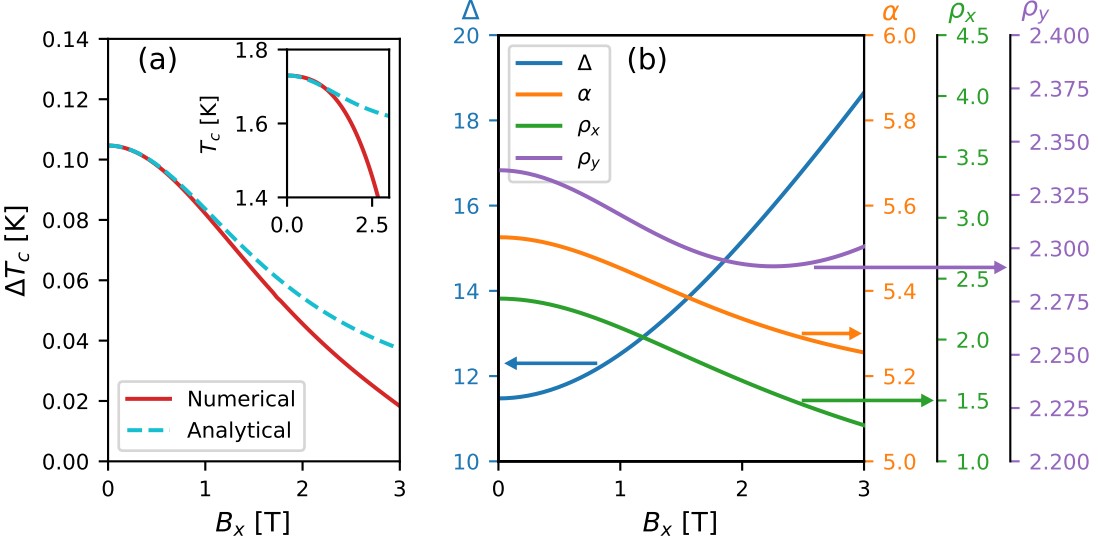

Figure 7: Dependence of phase transition parameters on external transverse magnetic field $B_x$. (a) plots the shift in critical temperature due to ODD terms $\Delta T_c$ with a comparison between numerical results and the analytical prediction of Eq. (31); the inset shows how the analytical prediction of Eq. (30) compares to the numerical result for $T_c(B_x)$. (b) illustrates the variation of the different parameters defined in Eq. (29) as functions of $B_x$, providing insights into their individual contributions to the behavior of $\Delta T_c$.

To summarize, though the approach presented in this section is limited to small fields, and is superseded by the semi-numerical approach described in Section 4, its significance is in providing some analytical insight, albeit approximate, into the $B_x > 0$ regime, clarifying the source of the steep rise of the phase boundary.

## G    Simulation details

The Hamiltonian used in MC simulations is very similar to $H_{\text{eff}}(B_x = 0)$ given by Eq. (5) in the main text, with two exceptions. The first, described in the main text, is that quantum terms, i.e., terms involving $\sigma^x$ or $\sigma^y$ operators, are omitted. The second is that the effective Hamiltonian is derived from the 17-state model rather than the three-state model.

Since the quantum terms have a vanishing MF contribution, the Hamiltonian used in MC simulations is essentially the same as the one used to obtain the MF results shown in Fig. 2 in the main text. The two differences introduced to the classical terms by switching to the 17-state model are in the magnitudes of the coefficients and in the emergence of a secondary, diagonal, three-body interaction of magnitude $\Gamma_D$, so

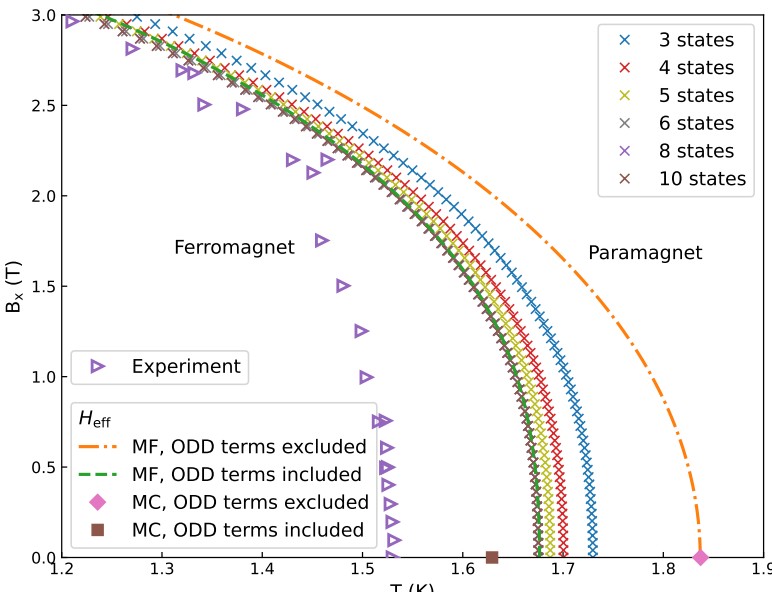

Figure 8: A subset of the $B_x - T$ phase diagram of LiHoF$_4$ illustrating the effect of including more excited states in the derivation of the effective Hamiltonian. X's of different colors represent MF results of $H_{\text{eff}}$ with a different number of states considered. The three-state model, described in detail in the main text, shows the smallest reduction, while, evidently, a 6-state model already sufficiently captures the full effect shown by the 17-state model used in this work (indicated by a dashed green line).

$$
\begin{aligned}
H_{\text{MC}} =& \frac{\alpha^2}{2} E_D \sum_{i \neq j} V_{ij}^{zz} \sigma_i^z \sigma_j^z + \frac{\alpha^2}{2} J_{\text{ex}} \sum_{i \neq j} \sigma_i^z \sigma_j^z \\
& - \Gamma E_D^2 \sum_{i \neq j \neq k} \left( V_{ik}^{xz} V_{jk}^{xz} + V_{ik}^{yz} V_{jk}^{yz} \right) \sigma_i^z \sigma_j^z \\
& - \Gamma_D E_D^2 \sum_{i \neq j \neq k} V_{ik}^{zz} V_{jk}^{zz} \sigma_i^z \sigma_j^z \\
& + \sum_{i \neq j} \tilde{\varepsilon}_{ij}^{zz} \sigma_i^z \sigma_j^z
\end{aligned}
\tag{32}
$$

where

$$
\Gamma = \alpha^2 \sum_{i=3}^{17} \frac{\langle i | J^x | \uparrow \rangle \langle \uparrow | J^x | i \rangle}{\langle i | V_c | i \rangle - \langle \uparrow | V_c | \uparrow \rangle} \approx 1.5 \frac{\alpha^2 \rho^2}{\Delta}
$$

$$
\tilde{\varepsilon}_{ij}^{zz} = 1.623 \frac{\varepsilon_{ij}^{zz}}{\rho^4 / 2\Delta} \approx 1.25 \varepsilon_{ij}^{zz}
$$

$$
\Gamma_D \approx 0.27 \, \text{K}^{-1}
$$

Here we see that the inclusion of all 17 crystal-field states increases the magnitudes of the emergent three-body and two-body interactions by about 50% and 25%, respectively, compared with the inclusion of just one excited state. The cumulative effect of including higher

excited states is demonstrated in Fig. 8. The secondary three-body interaction, of magnitude $\Gamma_D$, is almost negligible compared to the other three-body terms. In the simulation, periodic boundary conditions are used, with the long-range interactions handled using the Ewald summation method [61, 62].

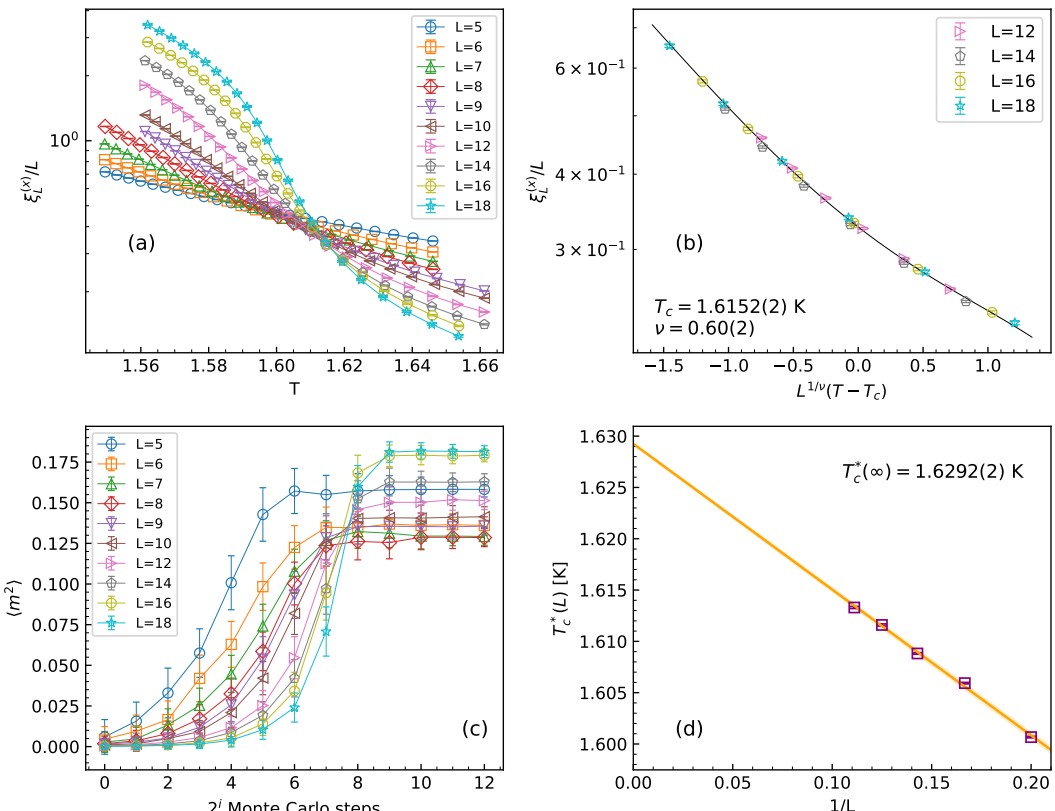

Figure 9: Monte Carlo data for LiHoF$_4$ at $B_x = 0$. (a) Finite-size correlation length divided by linear system size $L$ vs. temperature $T$. The clearly visible crossing indicates a phase transition. (b) Finite-size scaling analysis of the four largest system sizes, showing that, for an appropriate choice of parameters, data for different system sizes all fall on the same curve, indicated by the solid line (polynomial approximation). (c) Equilibration process of $\langle m^2(0) \rangle$ at $T \approx 1.58$ K. The $i$-th data point contains the average of $2^i$ MC sweeps, also averaged over independent simulation runs. (d) Extrapolation of $T_c$ to the thermodynamic limit. Each data point represents an estimation of $T_c$ obtained by finite-size scaling analysis of system sizes $L$ and $2L$. A linear fit of the data gives $T_c(\infty) = 1.6295(2)$ K.

## Equilibration and finite-size analysis

We use the parallel tempering Monte Carlo method [63]. To determine the critical temperature, we use the finite-size correlation length [64, 65],

$$\xi_L = \frac{1}{2\sin(k_{\min}/2)} \left[ \frac{\langle m^2(0) \rangle}{\langle m^2(\boldsymbol{k}_{\min}) \rangle} - 1 \right]^{\frac{1}{2}} \tag{33}$$

where

$$m(\mathbf{k}) = \frac{1}{N} \sum_{i=1}^{N} \sigma_i^z \exp(-i\mathbf{k} \cdot \mathbf{R}_i). \tag{34}$$

Here $\langle . \rangle$ refers to a thermal (MC) average. $\mathbf{R}_i$ is the location of the site $i$ and $\mathbf{k}_{\min} = \left(\frac{2\pi}{L}, 0, 0\right)$. The finite-size correlation length divided by the linear system size $L$ has a known scaling form,

$$\frac{\xi_L}{L} \sim \tilde{X}\left(L^{1/\nu}(T - T_c)\right) \tag{35}$$

so that for $T = T_c$, it is independent of the system size $L$, and curves of different sizes should cross. Assuming the scaling function (35) is well approximated by a third-order polynomial close to the critical point, we perform a non-linear fit for the four polynomial coefficients, $\nu$, and $T_c$. The crossing and fitting procedure are illustrated in Fig. 9(a) and Fig. 9(b), respectively. Statistical errors are estimated using the bootstrap method [66]. Finally, to extrapolate to infinite systems, we perform the above analysis for pairs of distinct system sizes $(L, 2L)$ and evaluate the crossing temperature $T_c^*(L, 2L)$ from each. The critical temperature in the thermodynamic limit is determined as the crossing of a linear fit to the $T_c^*(L, 2L)$ vs. $1/L$ data with the vertical axis, as shown in Fig. 9(d).

Equilibration of the simulation is verified by logarithmic binning of the data, i.e., the simulation time in terms of MC sweeps is successively increased by a factor of 2, and observables are averaged over that time. Once all observables of interest in three consecutive bins agree within error bars, the simulation is deemed equilibrated [66]. Fig. 9(c) shows the equilibration process of $\langle m^2(0) \rangle$. We run $N = 100$ independent simulations, each for $2^{12}$ MC sweeps for equilibration followed by another $2^{12}$ MC sweeps for measurement.

## H    Effective low-energy description of the Fe$_8$ molecular magnet

As described in the main text, the Fe$_8$ crystal consists of large molecular clusters, each described by a simple $S = 10$ spin model subject to a strong uniaxial anisotropy [3,42,46]. It is therefore well-described by the following Hamiltonian,

$$V_{\text{Fe}_8} = -DS_z^2 + E\left(S_x^2 - S_y^2\right) \tag{36}$$

where $D/k_B = 0.294\,\text{K}$ and $E/k_B = 0.046\,\text{K}$ [42], defining the hard, medium, and easy axes. The structure of the Fe$_8$ crystal is triclinic with $a = 10.676\,\text{Å}$, $b = 14.113\,\text{Å}$, and $c = 15.147\,\text{Å}$, and with $\alpha = 89.45°$, $\beta = 109.96°$, and $\gamma = 109.03°$ [46]. The orientation of the magnetic anisotropy axes is determined in Ref. [46] and used in this work. It gives a magnetic easy-axis that creates an angle of around $15°$ with the crystallographic $a$-axis.

Projecting $V_{\text{Fe}_8}$ and the angular momentum operators onto the four lowest-energy states of $V_{\text{Fe}_8}$, we get

$$V_{\text{Fe}_8} = \begin{pmatrix} 0 & & & \\ & 0 & & \\ & & \Omega & \\ & & & \Omega \end{pmatrix}; \quad J^z = \begin{pmatrix} -\beta & & & \\ & \beta & & \\ & & -\gamma & \\ & & & \gamma \end{pmatrix};$$

$$J^x = \begin{pmatrix} 0 & 0 & \sigma & 0 \\ 0 & 0 & 0 & -\sigma \\ \sigma & 0 & 0 & 0 \\ 0 & -\sigma & 0 & 0 \end{pmatrix}; \quad J^y = \begin{pmatrix} 0 & 0 & -i\chi & 0 \\ 0 & 0 & 0 & -i\chi \\ i\chi & 0 & 0 & 0 \\ 0 & i\chi & 0 & 0 \end{pmatrix} \tag{37}$$

where the states are chosen such that $\sigma = 2.06$, $\chi = 2.43$, $\beta = 9.99$, and $\gamma = 8.97$ are real and $\Omega = 5.51\,\text{K}$. The only non-negligible interaction between the $Fe_8$ molecular clusters is dipolar [45], so in the absence of an external transverse field

$$\mathcal{H}_{\text{Fe}_8} = \sum_i V_{\text{Fe}_8} + \frac{1}{2}E_{DF}\sum_{\substack{i\neq j \\ \nu,\mu}} V_{ij}^{\nu\mu} J_i^{\nu} J_j^{\mu}$$

with $E_{DF} = \frac{\mu_0 \mu_B^2 g^2}{4\pi}$ where $g = 2$ [42]. Applying the Schrieffer-Wolff transformation to obtain an effective low-energy spin-$\frac{1}{2}$ Hamiltonian, and then using a mean-field approximation, as before, gives a single self-consistency equation that yields

$$
\begin{aligned}
T_c =& -\beta^2 E_{DF}\sum_{j(\neq i)} V_{ij}^{zz} + \frac{2\beta^2\sigma^2}{\Omega}E_{DF}^2\sum_{j\neq k(\neq i)} V_{ij}^{xz}V_{jk}^{xz} \\
&+ \frac{2\beta^2\chi^2}{\Omega}E_{DF}^2\sum_{j\neq k(\neq i)} V_{ij}^{yz}V_{jk}^{yz} \\
&+ \frac{\sigma^2\chi^2}{\Omega}E_{DF}^2\sum_{j(\neq i)}\left(\left(V_{ij}^{xy}\right)^2 - V_{ij}^{xx}V_{ij}^{yy}\right) \\
=& -\beta^2 E_{DF}A_{zz} + \frac{2\beta^2\sigma^2}{\Omega}E_{DF}^2\left(A_{xz}A_{xz} - B_{xz,xz}\right) \\
&+ \frac{2\beta^2\chi^2}{\Omega}E_{DF}^2\left(A_{yz}A_{yz} - B_{yz,yz}\right) \\
&+ \frac{\sigma^2\chi^2}{\Omega}E_{DF}^2\sum_{j(\neq i)}\left(B_{xy,xy} - B_{xx,yy}\right) \\
=& \,0.95\,\text{K}.
\end{aligned}
$$

The $A_{\mu\nu}$'s and $B_{\mu\nu,\sigma\tau}$'s are calculated using Eq. (22). Indeed, the contribution of all but the first term above gives a negligible $-0.06\,\text{mK}$, consistent with the lack of an observable discrepancy between MF and experiment.

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
