# Peer review of "LiHoF4 as a spin-half non-standard quantum Ising system"

_SciPost Physics_

## Round 2 · Referee Report · Anonymous (Referee 1) · 2024-2-16

Strengths
-
The paper addresses an interesting question, viz. why do previous theoretical predictions for the critical temperature curve of LiHoF4 in a transverse field have a different shape from the actual experimental curve in the low-field limit?
-
The paper provides a convincing (partial) answer to that question, viz. that fluctuations out of the two-state low-energy manifold induce extra terms in the dipolar interaction that become especially effective at suppressing Tc when the transverse field is low.
-
The paper highlights the importance of the crystal structure by demonstrating that, although similar effects are present in other Ising-type systems such as Fe8, they are quantitatively much smaller.
Weaknesses
-
The answer given to the question is not new - the key physical idea was presented already in ref. [30], though admittedly here the authors have managed to extend at least some parts of their analysis to arbitrarily large transverse field, which was not done in ref. [30] (or, as far as I know, anywhere else).
-
The methodology is an uncomfortable (to me) mix of unbiased numerical treatment and ad hoc phenomenological renormalisations. Examples of the latter are the rescaling of the longitudinal interaction and the temperature-dependent renormalisation of the transverse field to account for the effect of hyperfine interactions (see text at top of page 7). Even worse, data with two different values of the longitudinal-interaction scaling factor are presented in the paper.
-
There appear to be inconsistencies in notation, which make the argument hard to follow. For example, the Hamiltonian that includes all 17 crystal-field states on each site is labelled H_{full} on page 3, referred to in the text as "the full microscopic Hamiltonian," and labelled H_{micro} in Fig. 2. (At least, I think after a couple of readings that H_{full} and H_{micro} are the same thing; if they're not, the authors should explain clearly how they differ from each other.)
-
There appear to be mistakes in the specification of the model. For example, the text of page 4 says that H_{3B} given in (6) "gathers three-body terms," but the first term on the right-hand side of (6) appears to be a two-body interaction.
-
In some places the notation is unclear, e.g. there should be a site label i on the crystal-field term in (1) (otherwise what is being summed over?).
Report
As requested, I assess the paper here against the journal's acceptance criteria. My overall assessment is that one of the four 'expectations' criteria is met, and although some of the 'general acceptance' criteria are not, the weaknesses are minor and remediable. I therefore recommend publication of the paper in SciPost Physics after the authors have addressed the issues identified below.
Details:
Expectations (at least one required) - the paper must:
Detail a groundbreaking theoretical/experimental/computational discovery;
There is certainly a theoretical/computational discovery here, which is that including the ODD terms in the effective Hamiltonian for LiHoF4 flattens off the critical temperature as a function of transverse field at low transverse fields, making it look more like the experimental result. The suggestion that this would happen was made in their ref. [30], but this paper provides evidence - within the limits of their mean-field and Monte Carlo treatments - that it actually does. "Groundbreaking" is of course a subjective term, but I would say that this is of a level of significance comparable with other things that I've seen published in SciPost Physics.
Present a breakthrough on a previously-identified and long-standing research stumbling block;
Not clear; if there is a breakthrough, it's more the suggestion in ref. [30] that ODD terms could be having an important effective than the detailed elaboration on that suggestion presented here. It's also not clear to me that the discrepancy between experiment and theory near zero transverse field in this material would count as "a long-standing research stumbling block."
Open a new pathway in an existing or a new research direction, with clear potential for multipronged follow-up work;
Not really: one can imagine follow-up work that refined the effect of the ODD terms, took the Schrieffer-Wolff transformation to higher orders, etc., but not to the level of innovation that one would describe as "open[ing] a new pathway".
Provide a novel and synergetic link between different research areas.
No.
General acceptance criteria (all required) - the paper must:
Be written in a clear and intelligible way, free of unnecessary jargon, ambiguities and misrepresentations;
At the moment it doesn't quite satisfy this, I would say: too much important stuff (e.g. the parameter value of J_{ex} used when generating Fig. 2, or arguments justifying the choice of scaling factor of the longitudinal interaction) is referred to briefly or left for the reader to find in the references. See also the note above about apparent errors in the specification of the model and unclarities in the notation.
Contain a detailed abstract and introduction explaining the context of the problem and objectively summarizing the achievements;
Yes, this is done well.
Provide sufficient details (inside the bulk sections or in appendices) so that arguments and derivations can be reproduced by qualified experts;
Not quite - as far as I could see, not all parameter values are given, and one has to dig quite a way into the references to find the exact form of the temperature-dependent renormalisation of the transverse field used to account for the effect of hyperfine interactions.
Provide citations to relevant literature in a way that is as representative and complete as possible;
Yes, this is done well.
Provide (directly in appendices, or via links to external repositories) all reproducibility-enabling resources: explicit details of experimental protocols, datasets and processing methods, or processed data and code snippets used to produce figures, etc.;
Not quite - see comment above about parameters. More detail on the Monte Carlo simulation would also be useful. Was a stock code used, or was it home-grown? If the latter, is it available somewhere in case the reader wants to reproduce the graphs?
Contain a clear conclusion summarizing the results (with objective statements on their reach and limitations) and offering perspectives for future work.
Yes, this is done well.
Requested changes
-
Carefully check the formulas, correcting apparent errors (e.g. the presence of a two-body interaction in H_{3B}) and ambiguities of notation (e.g. the lack of a site index on the crystal field terms and the apparent confusion between H_{full} and H_{micro}).
-
Expand on the content and justification of the phenomenological scaling and renormalisation procedures, indicating clearly how the results would be different if they were not included (and ideally presenting results without the scalings / renormalisations implemented, so that the reader can see their effects).
-
Ensure that sufficient information, including all parameters (e.g. J_{ex}), is provided for the reader to reproduce the results presented. Give references for the numbers quoted, e.g. the values of alpha and rho given below (4).
-
Discuss the validity of the truncation of the Schrieffer-Wolff transformation. (This arises especially because of the observation on page 4 that the three-body terms are long-range because of the additional summations. Doesn't that raise the issue that higher orders in the Schrieffer-Wolff transformation might yield four-, five-, etc.-body interactions that are equally important?)

---

## Round 2 · Referee Report · Anonymous (Referee 2) · 2024-3-18

Strengths
1) The paper addressed a persistent theory/experiment disagreement in the critical temperature of LiHoF4 in the limit of zero transverse field 2) Novel 3-spin terms were derived and shown to be comparable to other small residual couplings known to be present in addition to the dominant dipole-dipole interaction; These additional couplings appear to suppress the critical temperature albeit not as much as observed experimentally 3) The authors make a case that the magnitude of the effect depends strongly on material details (e.g. lattice structure) and illustrate that such effects are negligible for another somewhat similar magnet Fe8
Report
The manuscript under review is a detailed attempt to resolve the longstanding quantitative disagreement between theory and experiment in LiHoF4. The key new ingredient are the perturbative corrections to the bare dipolar coupling that are down by the ratio of dipole interaction to the gap to excited states that appear as 3-spin interactions -- these are small but apparently sufficient, based on authors' treatment within mean-field theory and monte-carlo.
1) I would have liked a better summary justifying various rescaling factors invoked (with refs. to experimental works and "strong c-axis fluctuations" -- i do not recall the issue and not inclined to go digging in the literature), esp. considering the paper is all about fixing relatively small discrepancies
2) it might have helped to state the symmetry constraints on the 3-body terms, e.g. I assume they are required to be even in \sigma_z operators so as not to break two-fold symmetry.
These would limit the first of these terms, which apparently only contains 2 Pauli operators (both \sigma_z^s), to be a two-spin operator and thus be considered properly as E_D/\Delta order reduction of the bare 2-spin dipolar term rather than a 3-spin term. if this is the case, the authors should not call it a 3-spin term AND also address whether this term by itself (without the true 3-spin terms) can reproduced the reduction of the critical temperature or not.
3) it seems the 3-spin terms in the Hamiltonian imply the existence of non-classical expectation values induced by classical 2-point correlators, i.e. finite values of <X> and <Z> (possibly with shape-dependent spatial profiles?). If true, the authors may want to discuss these as possible predictions for future experiments.
Requested changes
1) better justfiication of rescaling factors
2) symmetry constraints on and possibly correction/reclassification of 3-spin terms

---

## Editorial Decision

unknown